# MOVING OUT: PHYSICALLY-GROUNDED HUMAN-AI COLLABORATION

## ABSTRACT

The ability to adapt to physical actions and constraints in an environment is crucial for embodied agents (e.g., robots) to effectively collaborate with humans. Such physically grounded human-AI collaboration must account for the increased complexity of the continuous state-action space and constrained dynamics caused by physical constraints. In this paper, we introduce *Moving Out*, a new human-AI collaboration benchmark that resembles a wide range of collaboration modes affected by physical attributes and constraints, such as moving heavy items together and maintaining consistent actions to move a big item around a corner. Using Moving Out, we designed two tasks and collected human-human interaction data to evaluate models' abilities to adapt to diverse human behaviors and unseen physical attributes. To address the challenges in physical environments, we propose a novel method, BASS (Behavior Augmentation, Simulation, and Selection), to enhance the diversity of agents and their understanding of the outcome of actions. Our experiments show that BASS outperforms state-of-the-art models in AI-AI and human-AI collaboration.

## 1 INTRODUCTION

Humans can quickly adapt their actions to physical attributes (e.g., sizes, shapes, weights, etc.) or constraints (e.g., moving with stronger forces, navigating narrow paths, etc) when collaborating with other agents in the physical world. This ability is critical when embodied agents (e.g., robots) need to collaborate with humans to complete real-world tasks, such as assembly, transporting items, cooking, and cleaning. In these scenarios, successful interactions require understanding physical attributes and constraints while aligning with human behavior.

Prior work (Carroll et al., 2019; Ng et al., 2022; Papoudakis et al., 2021; Puig et al., 2023; Christianos et al., 2020; Du et al.) has explored human-AI collaboration at the discrete/symbolic space or task level, which often has simplified interaction dynamics compared to the interactions in a physical world. In physically grounded settings, agents operate in a continuous state–action space where object interactions, motions, and task outcomes are affected by physical attributes and constraints such as mass, friction, shape, and contact dynamics. As shown in Fig. 1, physically grounded task settings have increased diversity of physical constraints, physical varia-

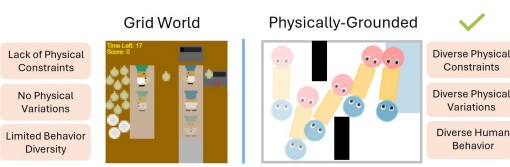

Figure 1: Multiagent collaboration in a grid world (Overcooked-AI (Carroll et al., 2019)) vs. in a physical world. Physically grounded settings introduce diverse physical constraints, attributes, and continuous low-level actions, which are essential for developing collaborative AI that can operate in physical scenarios.

tions, and human behavior in a continuous state-action space. While physical constraints, e.g., narrow passages, restrict movement and require precise coordination, there are still a large number of rotations or ways of holding objects that can lead to successful collaborations. In this paper, we propose *Moving Out*, a novel benchmark inspired by the Moving Out game (SMG Studio, 2020), to address physical interactions and diverse collaboration scenarios in a physically grounded setting.

While AI-AI collaboration can achieve strong collaborative performance through methods like self-play (Tesauro, 1994), the resulting AI agents often struggle to adapt to human-AI collaboration,

where human partners exhibit diverse behaviors (Carroll et al., 2019). This is particularly pronounced in physically grounded settings where minor variations in human actions, such as rotation angles or applied forces, can significantly affect outcomes. An agent needs to understand the physical consequences of actions to generalize behavior across different scenarios.

We design two tasks to evaluate an agent's ability to adapt to diverse human behavior and to understand physical constraints. The first requires the agent to play against unseen human behavior. We collected over 1,000 pairs of human demonstrations on maps with fixed physical properties from 36 human participants. These demonstrations capture a wide range of behaviors for identical set of tasks. The second requires the agent to generalize to unseen physical attributes and constraints. We collected 700 pairs of demonstrations from 4 experts on maps with random sampled object properties, such as mass, size, and shape. Together, these tasks provide a framework for testing the adaptability and generalization of embodied agents in diverse, physically grounded settings.

To further address the challenges of diverse behavior in the continuous state-action space and constrained transitions in physical environments, we propose BASS (Behavior Augmentation, Simulation, and Selection), a novel human-AI collaboration model which significantly outperforms prior works. First, we design a behavior augmentation strategy to enhance the diversity of the agent's collaborative partners. When an agent's start and end poses in one sub-trajectory match the sub-trajectory in another interaction, we can swap the partner's states to create new trajectories. This enables the agent to generate consistent behavior when the partner's behavior has variations. Second, we train a dynamics model of agent interactions so we can simulate the outcome of an action for a given state while considering the possible partner actions. We use the predicted states to score action candidates, allowing the agent to select actions that are more effective given the physical constraints. We evaluate BASS on the two proposed tasks in AI-AI and human-AI collaboration settings. We show that BASS outperforms baselines across key metrics such as task completion and waiting time. Our user study evaluated the model's performance against human participants, demonstrating the effectiveness of BASS in coordinating and assisting real humans.

In summary, our work makes the following contributions: (1) We introduce *Moving Out*, a continuous environment for physically grounded human-AI collaboration. (2) We propose two tasks and collect a human dataset to examine how human behavior and physical constraints impact collaboration. It is the first benchmark with human-collected dataset designed to study continuous, low-level motion control. (3) We develop *Behavior Augmentation, Simulation, and Selection* (BASS), which significantly improves human-AI collaborative performance in physically grounded settings.

## 2 RELATED WORK

**Multi-Agent Environments for Human-AI Collaboration** Several multi-agent environments (Leibo et al., 2021; Terry & Black, 2020) have been proposed for multi-agent reinforcement learning (MARL), but many are competitive rather than cooperative. For human-AI collaboration, prior environments largely adopt symbolic or discrete action spaces, such as OvercookedAI (Carroll et al., 2019), LBF and RWARE (Christianos et al., 2020), Hanabi (Bard et al., 2020), or social settings like Watch and Help (Puig et al., 2020) and Smart Help (Cao et al., 2024). While these settings are useful for studying coordination, they lack rich physical constraints and embodied teamwork. Other efforts, including It Takes Two (Ng et al., 2022), HumanTHOR (Wang et al., 2024a), and Habitat 3.0 (Puig et al., 2023), incorporate more realistic simulation. However, It Takes Two provides only a single, highly simplified task, while HumanTHOR and Habitat 3.0 focus primarily on navigation or high-level task coordination. In contrast, Moving Out provides continuous control, diverse physical attributes, and multiple collaboration modes, enabling the study of how AI can adapt to human behaviors under physical constraints. For a summarized comparison, see Appx. A.

**Learning Human-AI Collaboration Policy** Behavior Cloning (BC) is a common paradigm for learning policies from human demonstrations, typically using MLPs (Rumelhart et al., 1985), GRUs (Cho et al., 2014), or diffusion models (Chi et al., 2023). Beyond BC, several works extend imitation by predicting and scoring future states or trajectories, such as future-state prediction (Wang et al., 2024b; Kang & Kuo, 2025), interactive agent forecasting (Yuan et al., 2021), and trajectory-level scoring (Zhao et al., 2021; Kobayashis, 2020). Reinforcement learning (RL) approaches further enhance collaboration via self-play (Tesauro, 1994) and population-based training (Jaderberg et al., 2017), encouraging diverse behaviors for zero-shot coordination (Carroll et al., 2019; Strouse et al., 2021; Yan et al., 2023; Li et al., 2023; Yu et al., 2023; Zhao et al., 2023; Sarkar et al., 2023). Standard

multi-agent RL algorithms (Yu et al., 2022; Lowe et al., 2017), have also been applied. However, most RL methods rely solely on self-play without human data; more recent work integrates BC-trained models into the RL loop to align agents with human behavior (Liang et al., 2024; Carroll et al., 2019).

**Evaluating Human-AI Collaboration**  Research on human-AI collaboration has focused on evaluating and improving AI agents across different settings. (Attig et al., 2024) define evaluation criteria beyond task performance, incorporating aspects like trust and perceived cooperativity . In AI-assisted decision-making, (Vollmuth et al., 2023) directly computes the accuracy of AI decisions. Some works (Tylkin et al., 2021; Strouse et al., 2021; Sarkar et al., 2023) focus on training RL agents to adapt to diverse partners and evaluate the agents by the score when playing with humans. Several works (Sarkar et al., 2023; Attig et al., 2024; Siu et al., 2021; McKee et al., 2024; Hoffman, 2019) design questionnaires to evaluate different aspects like human-like, trustworthiness, and fluency.

## 3  PROBLEM DEFINITION

We model human-AI collaboration as a decentralized Markov decision process (Dec-MDP) (Beynier et al., 2013; Boutilier, 1996), defined as $\mathcal{M} = (\mathcal{S}, \mathcal{A}, \mathcal{P}, r, \mathcal{O}, \gamma, T)$, where $\mathcal{S}$ is the joint state space, and $\mathcal{A} = \mathcal{A}^i \times \mathcal{A}^j$ is the joint action space of the two agents. The transition function $\mathcal{P} : \mathcal{S} \times \mathcal{A} \times \mathcal{S} \to [0, 1]$ is the probability of getting the next state given a current state and a joint action. The reward function $r : \mathcal{S} \times \mathcal{A} \to \mathbb{R}$ specifies the reward received for each state-joint-action pair. The observation function $\mathcal{O} : \mathcal{S} \to \mathcal{O}^i \times \mathcal{O}^j$ generates an observation for each agent for a given state. The observation of each agent makes the state jointly fully observable. The discount factor $\gamma \in [0, 1]$ determines the importance of future rewards, and $T$ is the time horizon of the task.

At each timestep $t$, the environment is in a state $s_t \in \mathcal{S}$. Agents $\pi^i$ observes $o_t^i \in \mathcal{O}$, where $\mathcal{O}$ is the observation space derived from $s_t$, and selects an action $a_t^i \in \mathcal{A}^i$ according to its policy $\pi^i : \mathcal{O} \to \mathcal{A}^i$. The joint action $a_t = (a_t^i, a_t^j)$ transitions the environment deterministically to a new state $s_{t+1} \sim \mathcal{P}(\cdot | s_t, a_t)$. The trajectory of an episode is defined as $\tau = (s_0, a_0, s_1, \dots, s_{T-1}, a_{T-1}, s_T)$, and the discounted return for the trajectory is: $R(\tau) = \sum_{t=0}^{T-1} \gamma^t r(s_t, a_t)$. The objective of each agent is to maximize the expected return $J(\pi^i, \pi^j) = \sum_\tau R(\tau)$ where the return is evaluated over the trajectories induced by the policies $(\pi^i, \pi^j)$.

**Challenges when Collaborating with Humans**  When one of the agents is a human, the human agent may have diverse behaviors (Carroll et al., 2019). The AI agent must adapt its policy $\pi^i$ to a wide range of potential human policies $\pi^j$. At inference time, we assume that the real human policy $\pi^j$ is drawn from a unknown human policy distribution $\mathcal{D}$. Thus, the AI agent's optimal policy is:

$$\pi_\star^i = \arg \max_{\pi^i} \mathbb{E}_{\pi^j \sim \mathcal{D}} \mathbb{E}_{\tau \sim (\pi^i, \pi^j)} [R(\tau)]$$

where $\mathbb{E}_{\tau \sim (\pi^i, \pi^j)}$ denotes the expectation over $\tau$ where the actions are drawn from $\pi^i$ and $\pi^j$ respectively. Since the ground-truth distribution $\mathcal{D}$ is unknown, the AI must use limited data to generalize across diverse human strategies.

The physical embodiment of agents and the physical environment introduce significant challenges for this human-AI collaboration framework. First, the continuous variables, e.g., positions and directions, increase the number of configurations in the state space. For example, there are multiple configurations that an agent can take to rotate an object together. The AI agent must optimize its policy under diverse human behaviors while ensuring robustness across a continuous and high-dimensional state space. Second, the state space $\mathcal{S}$ also includes continuous physical variables such as object positions, orientations, and attributes (e.g., shape, size, and mass), which can create several constraints to limit the feasible state transitions $\mathcal{P}$. For instance, when two agents jointly move an object, the physical properties of an object (e.g., mass or shape) can influence the required actions for successful transitions. Objects with irregular shapes require agents to coordinate their grips at specific parts. Heavier objects demand synchronized forces of two agents. Considering the physical constraints $\Gamma(s_t, a_t)$ that apply to the current state-action pair, the transition function is constrained as follows:

$$\mathcal{P}(s_{t+1} \mid s_t, a_t) = \begin{cases} 1, & \text{if } \Gamma(s_t, a_t) \text{ satisfies (transition to } s_{t+1}) \\ 0, & \text{if } \Gamma(s_t, a_t) \text{ does not satisfy (remains in } s_t) \end{cases}$$

These constraints create several narrow transitions, similar to prior studies about motion planning (Hsu et al., 2003; Saha et al., 2005; Szkandera et al., 2020), and can further affect the agents' collaboration

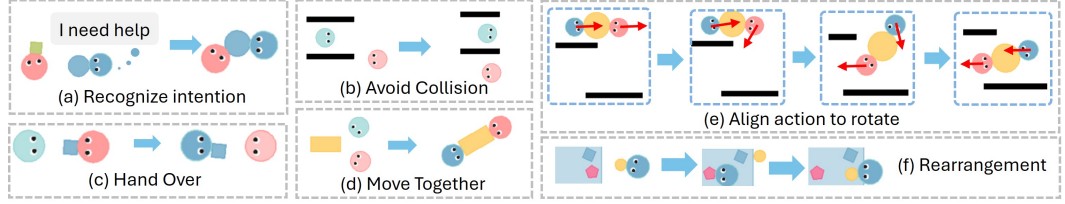

Figure 2: *Moving Out* requires two agents to collaboratively move objects to the blue goal regions. The environment includes movable objects with varying shapes and sizes. An agent can move a small item quickly. As the object sizes increase, the agent needs the other's help to move the object.

strategies. For example, in scenarios where the agents need to move a rectangular sofa through a narrow doorway, the agents need to grasp the shorter sides of the sofa and coordinate their moves to ensure they can fit through the entrance without collision. In this paper, we study human-AI collaboration under the challenges of continuous state space and constrained transitions introduced by physical embodiments and environments.

# 4 MOVING OUT ENVIRONMENT AND DATASET

## 4.1 ENVIRONMENT

To test how physical environments can affect human-AI collaboration, we need an environment that follows physics. We build *Moving Out* on top of a single-agent environment Magical (Toyer et al., 2020; Zakka et al., 2021) where agents and objects are physical bodies moving in a 2D physics simulation. Similar 2D physics engines have also been adopted in recent works studying physical reasoning and embodied AI (Morlans et al.; Li et al., 2024a; Liu et al., 2024). As shown in Fig. 2, each agent can maneuver freely in Moving Out and move objects with varying degrees of difficulty depending on the object size and shape. The goal is to transport all objects to the goal regions. This design emphasizes flexibility, allowing agents to act independently while also creating scenarios where collaboration is necessary for efficient task completion.

### 4.1.1 PHYSICAL VARIABLES

The environment includes these physical components: movable items, walls, and goal regions.

**Movable Items** are controlled by the following variables to introduce diverse physical interactions.
• **Shapes** include stars, polygons, and circles, each requiring unique grabbing and rotation strategies.
• **Sizes** range from small to large, each has increasing difficulty in moving, and can slow agent speed.
• **Mass** is varied for different items. This influences an agent's moving speed during transportation.

**Walls** introduce friction. Agents that collide with walls experience reduced moving speed, adding another layer of complexity.

**Goal regions** are designated areas larger than the total size of items. Agents must carefully arrange items to ensure all items can fit in the region, requiring precise spatial planning and coordination.

Figure 3: Diverse collaboration behaviors in *Moving Out*, including (a) recognizing when help is needed, (b) avoiding collisions, (c) passing objects, (d) moving items together, (e) aligning actions, and (f) organizing objects in the goal region.

### 4.1.2 LAYOUT TYPES

The physical variables introduce diverse collaborative behavior as illustrated in Fig. 3. A successful collaboration usually requires a mixture of different behaviors. To systematically understand the collaborative performance of AI agents, we designed 12 maps focusing on three collaboration modes. See example maps in Fig. 4 and Appx. V for the full set of maps.

**Coordination**  The maps in this category only include small items, so each agent can complete the task independently. However, narrow passages in the maps often block an agent's path, requiring the partner to step aside or help pass the item. For example, in Map 1 (Hand Off), the blue agent must pick up the item and, because of the narrow passage, pass it to the pink agent. This setup enforces cooperation, as the task cannot be completed without coordination between the two agents.

**Awareness**  The maps in this category do not have a clear optimal sequence for moving items, requiring agents to decide whether, when, and how to assist their partner for efficiency.

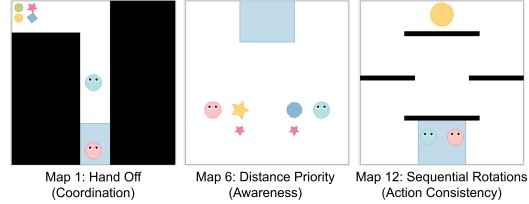

Map 1: Hand Off (Coordination)  Map 6: Distance Priority (Awareness)  Map 12: Sequential Rotations (Action Consistency)

Figure 4: Example maps in *Moving Out* focusing on different collaboration modes: coordination, awareness, and action consistency.

For instance, in Map 6 (Distance Priority), each agent starts near multiple items and must decide whether to handle nearby items first, assist their partner, or prioritize tasks independently. These decisions become even more complex when collaborating with a human partner, as human behavior can vary significantly. A human partner might wait for AI help with larger items, be passive, or focus on smaller tasks independently. This variability demands that the AI agent dynamically adapts to the human's behavior. In addition, these maps may involve *implicit communication* through physical actions. For example, slow movement when lifting a heavy object or a partner waiting for a long time can show that help is needed. This form of communication comes directly from the task dynamics and does not require extra language messages, making it possible to study human–AI collaboration without adding language or symbolic messages.

**Action Consistency**  This scenario requires agents to maintain consistent and synchronized actions over time, such as continuously aligning their efforts to move and rotate large items together. The challenge is aligning force directions and dynamically adjusting them to ensure efficient movement while navigating around tight spaces or obstacles. For instance, in Map 12 (Sequential Rotations), two agents must collaboratively transport a large item through a series of narrow passages. Throughout this process, the agents must continually synchronize their actions to rotate and adjust the item's angle, allowing it to fit through the openings. Misalignment in their efforts could result in the item becoming stuck or unnecessary movements that waste time and energy.

### 4.2 TASKS

We design two tasks that evaluate a model's ability to adapt to diverse human behaviors and to generalize to unseen physical attributes.

**Task 1: Adapting to Diverse Human Behaviors in Continuous Environments**  The first challenge of physically grounded human-AI collaboration arises from the continuous state-action space, which allows for a wide range of possible human behavior. To test whether an agent can adapt to diverse human behavior, we fixed the configurations of the 12 maps and collected human-human collaboration data that demonstrate different ways to collaborate in the same maps. These demonstrations represent a finite set of human behaviors. In this task, we train a model on this dataset and test it with a new human or AI collaborators. This setup assesses whether the model can generalize beyond the observed behaviors to adapt to diverse human behavior. For an agent designed to assist humans effectively, learning to adapt from limited human demonstrations is crucial.

*Task 1 Evaluation Protocol*  Simply training on the full dataset requires us to recruit human participants to play against the model during every test and can lead to highly variable results. To address the reproducibility issue, we split the dataset by participants into two disjoint splits, train separate AI agents on each split, and then evaluate them by letting the agents collaborate with each other. This protocol provides a reproducible proxy for testing generalization to unseen human behaviors.

**Task 2: Generalizing to Unseen Physical Constraints** The second challenge arises from the physical constraints, which limit the possible transitions of given states. To test whether the agent understands physical constraints, we randomized the physical attributes of objects in the 12 maps to collect human-human interaction data that demonstrates how humans adapt to changes in physical variables. Again, we train a model with the collected dataset and evaluate it on maps with unseen object attributes. To ensure the model learns the effects of physical constraints rather than memorizing them, we avoid having identical objects in the training and testing datasets. In particular, the variation is defined compositionally over the object's physical properties, ensuring that evaluation maps always include unseen combinations (e.g., a large star-shaped object is excluded from training whereas only small stars and large squares are present). This forces the model to understand the impact of shape and type, and generalize across varying physical configurations.

*Task 2 Evaluation Protocol* Although evaluating directly with humans is possible, a more reproducible and efficient approach is to train agents on the full dataset and then test them via AI-AI self-play. Since evaluation maps contain object attributes not seen during training, this setup directly measures an agent's ability to generalize to unseen physical constraints.

### 4.3 DATASET

The data collection was approved by the Institutional Review Board (IRB). Two human players control the agents with joysticks. The game ran at 10Hz, and on average, each map took around 30 seconds (or 300 time steps) to transport all items. See Appx. R for details.

For Task 1, we recruited 36 college students as participants and collected over 1,000 human-human demonstrations (2,000 action sequences in total) across 12 maps. This ensures that the dataset captures a wide range of human behaviors, providing sufficient diversity for training and testing the model's ability to generalize to unseen human strategies.

As shown in Table 1, we compare the diversity of our dataset against datasets collected by RL agents or experts using Dynamic Time Warping (DTW; mean and variance), entropy, and coverage distance, showing ours has the best diversity. This demonstrates the effectiveness of recruiting diverse participants for data collection. See Appx. Bfor further details, including trajectory visualizations.

| Dataset | DTW Mean (↑) | DTW Var (↑) | Avg. Entropy (KDE) (↑) | Coverage Distance (RBF) (↑) |
|---|---|---|---|---|
| Moving Out Task 1 | **7.013** | **6.065** | **0.888** | **0.899** |
| Expert dataset | 4.642 | 3.029 | 0.757 | 0.744 |
| RL agent collected data | 4.358 | 2.499 | 0.683 | 0.626 |

Table 1: Dataset diversity across different data collection methods. Our dataset achieves consistently higher diversity compared to expert and RL agent datasets.

For Task 2, we emphasize the randomized properties of objects rather than the variable behaviors. In this case, we used 4 expert players to collect 720 human-human demonstrations (1,440 action sequences in total), with 60 demonstrations per map. Each map included randomized object physical attributes, where pose, mass, and size were varied by up to 10%, while object types and shapes were randomized to be different from those used in evaluation. This setup allows us to assess the model's ability to generalize to unseen object attributes. Fig. 5 shows examples of two maps.

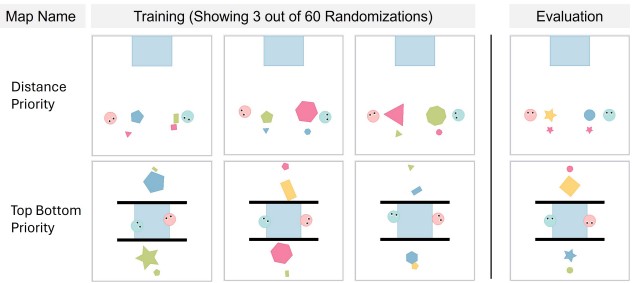

Figure 5: Randomization examples in Task 2, illustrating generalization to unseen physical properties.

## 5 BASS: BEHAVIOR AUGMENTATION, SIMULATION, AND SELECTION

To address the proposed tasks, we develop BASS (Behavior Augmentation, Simulation, and Selection) which considers the increased number of configurations in continuous space and the outcome of

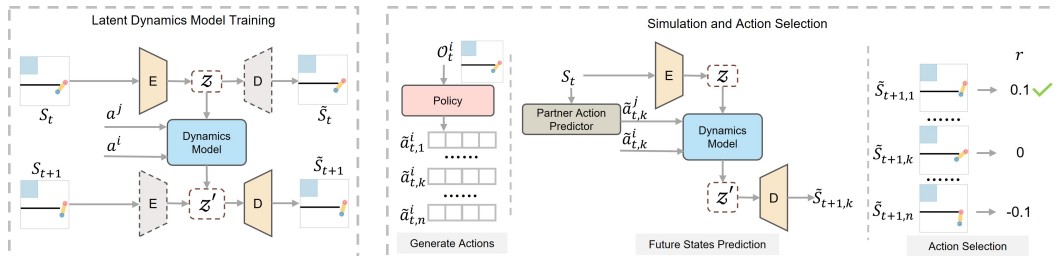

Figure 6: Overview of our Simulation and Action Selection components. **(Left)** The latent dynamics model that encodes the latent state from $t$ to $t + 1$ to enable next state prediction. **(Right)** The action selection pipeline: The policy first generates candidate actions. The dynamics model then estimates the resulting future states, and finally, the best action is selected based on state evaluation.

actions in physical environments. First, at training time, we augment the behavior data. This helps the model adapt to diverse behaviors better by exposing it to a broader range of possible interactions. Second, we train a dynamics model to simulate the outcome of an action, allowing the agent to understand the impact of actions on different physical properties. At inference time, the model select actions by evaluating the predicted states.

### 5.1 COLLABORATION BEHAVIOR AUGMENTATION

**Behavior Augmentation** Trajectory augmentation is already used in single-agent settings (Kim et al., 2024; Sussex et al., 2018), but extending this idea to multi-agent raises new challenges: naively altering one agent's behavior can easily break the consistency required for cooperation, since both agents must pursue aligned goals for the trajectory to remain valid. We adopt two strategies.

First, we perturb partner poses with small Gaussian noise, $\tilde{p}_{\text{partner}} = p_{\text{partner}} + \epsilon, \quad \epsilon \sim \mathcal{N}(0, \sigma^2)$, while keeping other state variables unchanged, where $p_{\text{partner}}$ is the original partner's pose, $\epsilon$ is Gaussian noise with mean 0 and variance $\sigma^2$, and $\tilde{p}_{\text{partner}}$ is the perturbed pose used to generate new state variations. This generates new states that mimic natural variations in human movement and improve robustness to small deviations.

Second, we augment the data by recombining sub-trajectories from two demonstrations. In successful demonstrations, if agent $i$'s sub-trajectories are the same, this indicates the corresponding agent $j$'s sub-trajectories are compatible with agent $i$'s, even if the ones from agent $j$ are very different. Taking this intuition, the key idea is to keep agent $i$'s behavior fixed while swapping the partner's sub-trajectories. We identify a segment of agent $i$ in demonstration A between timesteps $t_1$ and $t_2$, and another segment in demonstration B between $t_3$ and $t_4$, where the agent begins and ends in nearly the same state. Because of the continuous state space, we treat two states as equivalent when the difference in the agent's pose is below a very small tolerance $\epsilon_{\text{pose}}$, which is visually indistinguishable in practice. Formally, this matching condition is expressed as $s_{t_1}^i \approx \hat{s}_{t_3}^i$ and $s_{t_2}^i \approx \hat{s}_{t_4}^i$ with $t_2 > t_1$ and $t_4 > t_3$. Once these two segments match for agent $i$, we keep agent $i$'s motion unchanged and swap the corresponding partner $j$'s sub-trajectories between the two demonstrations. Further implementation details and a visualization example are provided in Appx. O.

Together, these augmentations expand the dataset with physically plausible trajectories that preserve collaboration-level coherence, a novelty compared to prior single-agent augmentation methods.

**Validity of the Augmented States** We can validate generated sub-trajectories based on the state information. Specifically, one can check whether the generated states remain within the valid state space and do not result in conflicts, such as collisions or other inconsistencies. However, as detailed in Appx. H, our recombination strategy is explicitly designed to ensure coherence (e.g., by swapping only sub-trajectories with identical start and end states). Even without additional validation, this augmentation strategy can improve performance, as we show in the experiments.

### 5.2 SIMULATION AND ACTION SELECTION

To understand the outcome of an action, in simulation environments, we can utilize the physics engine to simulate the action outcome. However, in real-world settings where a simulator is unavailable, a world model or next state predictor is required. Fig. 6 shows the training and inference pipelines of our Simulation and Action Selection components.

**Next State Prediction** Our next state predictor utilizes two autoencoders to estimate future states. First, one autoencoder encodes the current state into the latent space. The dynamics model then takes this latent representation along with the actions of both agents as input to predict the latent representation of the next state. Finally, this predicted latent representation is decoded by another autoencoder to reconstruct the next state. Since the next state depends on the agent's own action and the partner's action, we use a partner action predictor to estimate the partner's action based on the current state. Practically, the partner's predictor can share the same architecture as the agent's policy or directly use the agent's own policy by swapping its state with the partner's state to predict the partner's action. The dynamics model predicts the future state as: $z_{t+1} = f(z_t, a_t, a_t^{(p)})$, where $z_t$ and $z_{t+1}$ represent the latent spaces of the current and future states, $a_t$ is the agent's action, $a_t^{(p)}$ is the inferred partner's action, and $f$ is the dynamics model.

**Action Selection** Our policy and partner action predictor both require strong multi-modal modeling capacity to generate diverse action candidates, which forms the basis for action selection. Once the next state is predicted, the reward for each action is computed based on the total distance of all objects to the goal region. We use Normalized Final Distance (NFD) as defined in Sec. 6, but other metrics that measure partial progress of map completion also suffice. We then select the action with the highest reward as the optimal action: $a^* = \arg\max_{a_i} r(a_i), i = 1, 2, \ldots, n$, where $r(a_i)$ is the reward for action $a_i$. This approach enables the model to choose the most effective action, even in real-world scenarios without access to a simulator. A comparison of NFD against alternative objectives is provided in Appx. N.

## 6 EXPERIMENT

We aim to answer the following research questions: **(RQ1)** Does BASS adapt to unseen human behaviors with limited performance degradation? **(RQ2)** Does BASS generalize to unseen physical constraints? **(RQ3)** Does the multi-agent design in BASS effectively consider the partner's behavior? **(RQ4)** Does BASS work more effectively with humans in physically grounded collaboration? **(RQ5)** What failure patterns do existing methods and BASS exhibit?

To answer these questions, we train and test all methods on the two Moving Out tasks following the designed evaluation protocols, and then conduct a human study for further validation. For AI–AI collaboration, all results are averaged over 20 runs.

### 6.1 SETTINGS

**Baselines** We compare BASS against these behavior cloning and RL baselines to predict actions:

- **MLP** is a common behavior cloning baseline.
- **GRU** captures temporal dependencies of state and actions using recurrent connections.
- **Diffusion Policy (DP)** (Chi et al., 2023) captures multimodal distribution and has demonstrated strong performance across various tasks.
- **MAPPO** (Yu et al., 2022) is a commonly used multi-agent RL algorithm. It has demonstrated strong performance in cooperative games. See Appx. E for details about training.

**BASS Implementation** BASS builds on the same diffusion policy backbone used in our baselines, serving as both the base policy and the partner action predictor because of its strong multi-modal modeling capacity. The VAE and dynamics models are implemented as MLPs and co-trained. For action selection, the policy and partner predictor independently sample 4 action candidates each. While increasing the number of samples could further improve accuracy, collaboration requires real-time inference; sampling four candidates ensures inference can be performed at 10Hz. Ablation studies on the sampling strategy, analyses of individual modules are provided in Appx. K, J.3, L.1.

**Evaluation Metrics** We measure the success of collaboration using the following metrics: (1) Task Completion Rate (TCR) for successful item delivery; (2) Normalized Final Distance (NFD) for the distances between objects and the target, measuring partial progress; (3) Waiting Time (WT) for the amount of time an agent waits for assistance with large items; and 4) Action Consistency (AC) for the degree of force alignment when moving items jointly, indicating coordination efficiency. Detailed definitions are in Appx. I.

**Human Subject Study** Our study was approved by the IRB. We conducted a human subject study with 32 participants to evaluate BASS against the DP baseline in both tasks. Each participant played 32 maps in total, cooperating with each method in two rounds per map. After completing the first round, the participant and model switched to control the other agent. Upon finishing all maps, participants were given a questionnaire to capture subjective feedback. See Appx. S for details.

| Evaluation Protocol | Method | TCR (↑) | NFD (↑) | WT (↓) | AC (↑) |
|---|---|---|---|---|---|
| Seen Behaviors | MLP | 0.2126 | 0.2987 | 0.4896 | 0.8013 |
| | GRU | 0.2369 | 0.3011 | 0.4975 | 0.8151 |
| | MAPPO | 0.1929 | 0.3182 | 0.5766 | 0.8097 |
| | DP | 0.3233 | 0.5367 | 0.3789 | 0.8163 |
| | DP/BASS | **0.3503** | **0.5724** | **0.3598** | **0.8337** |
| Unseen Behaviors | MLP | 0.1433 (-32.61%) | 0.2413 (-19.22%) | 0.5647 (+15.33%) | 0.7729 (-3.54%) |
| | GRU | 0.1638 (-30.87%) | 0.2453 (-18.53%) | 0.5758 (+15.74%) | 0.7830 (-3.94%) |
| | MAPPO | 0.1635 (-15.19%) | 0.2808 (-11.74%) | 0.6379 (+10.64%) | 0.7858 (-2.95%) |
| | DP | 0.2563 (-20.72%) | 0.4589 (-14.50%) | 0.4249 (+12.15%) | 0.7854 (-3.78%) |
| | DP/BASS | **0.3010 (-14.07%)** | **0.5197 (-9.22%)** | **0.3899 (+8.37%)** | **0.8099 (-2.86%)** |
| Play with Human | DP | 0.3855 | 0.5547 | 0.4886 | 0.8054 |
| | DP/BASS | **0.6512** | **0.7053** | **0.3364** | **0.9124** |

Table 2: Task 1 results under seen and unseen human behaviors, and with real human partners.

## 6.2 RESULTS

**Collaboration with Unseen Behaviors (RQ1)** Table 2 reports Task 1 results under three protocols. In the seen setting, agents are trained and evaluated on the same dataset. Here, both DP and BASS outperform MLP, GRU, and MAPPO, with BASS achieving the best task completion (TCR, NFD). In the unseen setting, we split participants into disjoint sets as described in the evaluation protocol, train separate policies, and then evaluate them by playing across groups. All methods degrade when facing unseen behaviors, but BASS shows the least performance drop across TCR, NFD, WT, and AC, indicating stronger robustness. See Appx.T for the full table with standard error.

**Collaboration under Unseen Physical Constraints (RQ2)** Fig. 7 shows that, while waiting time and action consistency are comparable across methods, BASS consistently outperforms baselines, particularly in TCR and NFD, which directly reflect task progress under new object properties. This suggests that evaluating candidate actions based on predicted future states helps the model better adapt to variations in size, mass, and shape.

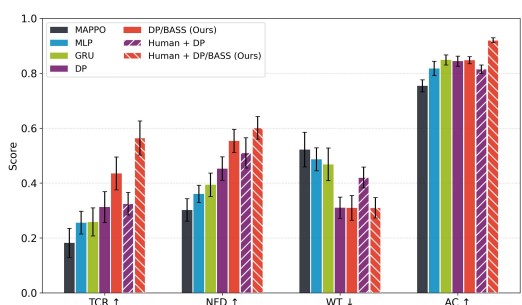

Figure 7: Results on Task 2 under unseen physical constraints.

**Effectiveness of the Multi-agent Design in BASS (RQ3)** To show the importance of modeling both agents, we compare BASS with a single-agent variant that ignores partner alignment during recombination and predicts only one agent's future state during action simulation. The single-agent versions increase diversity but noticeably reduce collaboration performance. For example, in Task 1, TCR and NFD drop from {0.403, 0.511} in the full multi-agent version to {0.368, 0.451} with single-agent recombination. In Task 2, the multi-agent action simulation achieves {0.420, 0.554}, whereas the single-agent variant reduces these to {0.319, 0.458}. These results show that a multi-agent model structure is necessary for generating valid augmented trajectories and selecting effective actions. Full results and tables are provided in Appx. L.2.

**Collaboration with Humans (RQ4)** Tab. 2 and Fig. 7 show the results with humans. In both tasks, BASS significantly improved task completion rates (TCR and NFD) compared to the DP. This indicates that BASS adapts better to human behavior. For wait time, DP increased when

playing with humans, suggesting it struggles with different humans, despite DP capturing multimodal distributions. BASS reduced wait time, demonstrating its ability to adapt to diverse behaviors. For action consistency, DP performed worse because it cannot handle differences between the evaluation and training data. BASS augmented diverse collaborative behaviors during training and selected the best actions for interacting with humans, resulting in better consistency.

**Human Feedback (RQ4)** Fig. 8 summarizes post-experiment survey results from humans. We compare BASS with DP. The results show that BASS significantly outperformed DP in the Helpfulness category, indicating that BASS is better at consciously assisting others. Additionally, BASS demonstrated a better understanding of physics, suggesting that our next state predictor effectively comprehends and evaluates different actions to choose the best ones. Independent t-tests revealed that these differences are statistically significant ($p = 0.017$).

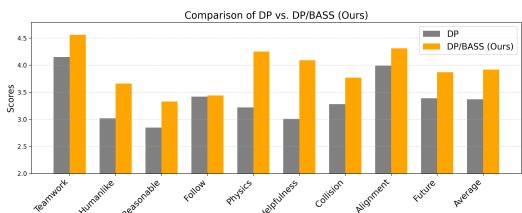

Figure 8: User survey results in a 7-point Likert scale

**Failure case study (RQ5)** Fig. 9 shows examples of common failure cases from DP. In task 1, as illustrated in failure case 1, many participants reported that the AI agent frequently holds an item without passing it, resulting in frequent collisions. Additionally, participants noted that the AI agent often failed to come to assist, as shown in failure case 2, where a human agent (blue) was slowly pulling an item, but the AI agent (pink) instead went to grasp other smaller objects. These issues show the model's limited ability to adapt to diverse behaviors, making it difficult to respond appropriately to actions that were not present in the training dataset.

In task 2, most participants pointed out failure case 3, where the AI agent reached the target item but was unable to successfully grasp it. This indicates that the model struggles when encountering objects that were not in the training data. In contrast, BASS shows fewer reported failure cases than DP. Manual inspection revealed that BASS reduced the occurrence rates of the three failure types from $\{0.797, 0.688, 0.906\}$ in DP to $\{0.343, 0.563, 0.484\}$. However, effectively addressing these failures remains a substantial challenge for future research.

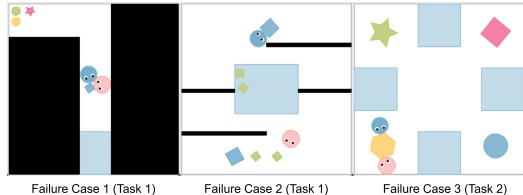

Figure 9: Failure cases: 1) Failing to release items during handover, 2) Not responding when assistance is needed, and 3) Inability to grasp large items upon approach.

# 7 CONCLUSION

We introduce *Moving Out*, a physically grounded human-AI collaboration benchmark that features a continuous state-action space and dynamic object interactions. We created two challenging tasks and collected human-human collaboration data to enable future model development. Our evaluation results show that much remains to be done with existing models to effectively collaborate with humans in physical environments. Our proposed method, BASS, takes the first step to improve models' adaptability to diverse human behaviors and physical constraints.

Future work includes strengthening the theoretical understanding of human–AI collaboration and the BASS framework, extending the benchmark toward richer and more complex cooperative scenarios, and adding explicit communication on top of the implicit communication already present in our current environment. These directions will enhance both the practical and theoretical aspects of physically grounded human–AI collaboration.

ETHICS STATEMENT

Our human subject study and data collection were approved by the Institutional Review Board (IRB), and all participants provided informed consent. We carefully removed personally identifiable or sensitive information before releasing the datasets. The study and dataset release strictly follow ethical guidelines for human subject research and data sharing.

REPRODUCIBILITY STATEMENT

We will release all source code, including the environment and model, to support reproducibility. The appendix includes detailed descriptions of the human study procedures to facilitate replication. Since human behaviors may vary and cannot be fully reproduced, we additionally provide reproducible AI-AI collaboration evaluation protocols for both Task 1 and Task 2, ensuring that our results can be independently verified.

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

## USE OF LARGE LANGUAGE MODELS

We used a large language model (LLM) (ChatGPT and Google Gemini) to support the writing of this paper. The usage was limited to the following purposes:

- **Polishing**: improving grammar, clarity, and flow of sentences.
- **Short Rewriting**: shortening paragraphs and sentences to save space while keeping the intended meaning clear.
- **Meaning Emphasis**: rewriting specific sentences to highlight or emphasize intended points.

Also, for coding support, the LLM provided assistance with minor coding tasks such as data pre-processing (e.g., converting file formats), introducing keyboard and joystick inputs, preparing data uploads to HuggingFace, and other small implementation details.

No part of the experimental design, analysis, or results was generated by the LLM.

# A  COMPARISON WITH OTHER ENVIRONMENTS

| Environment | State/Action | Physics-based | Constraints | Collaboration Behaviors |
|---|---|---|---|---|
| Overcooked-AI | Discrete | No | Items placed only in specific locations | Passing items, dividing tasks, and collision avoidance |
| Watch and Help | Discrete | Yes | Partial observability, diverse objects and goals | Goal inference, cooperative help |
| Smart Help | Discrete | Yes | Capability limits (weight, height, open/close/toggle), partial observability | Awareness: Goal + capability inference, bottleneck help, avoid unnecessary takeover |
| Table Carrying | Continuous | No | No physical feedback, task ends upon collision | Joint carrying (i.e., action consistency) |
| Moving Out | Continuous | Yes | Realistic physics, friction, collision feedback, diverse items with physical properties. | Coordination, Awareness of needing help, joint carrying (i.e., action consistency). |

| Metrics | Pros | Cons | Human Data |
|---|---|---|---|
| Number of cooked onions in a limited time | Small state/action space, fast training, human data available | Limited behavior variety, simple tasks | Yes |
| Success Rate, speedup, cumulative reward | 3D environment, diverse household tasks | No physical variations, high computational cost | Synthesized |
| Success Rate (Goal-conditioned), Helping Necessity/Rate, Episode/Success-weighted Length | 3D physics, diverse tasks | Predefined actions, high computational cost | No |
| Success rate, Completion time | Continuous actions | No physics in interactions, single task, no dataset | No |
| Task Completion Rate, Normalized Final Distance, Waiting Time, Action Consistency | Realistic physics, multiple collaboration modes, physics feedback, human dataset available | Require high-frequency actions for smooth collaboration. | Yes |

Table 3: Comparison between Moving Out, Overcooked-AI, and Table Carrying. Overall, Moving Out offers more diverse collaboration modes and physical constraints due to its physics-based environment.

# B  TASK 1 DATA DIVERSITY ANALYSIS

## B.1  VISUALIZATION OF DATASET IN TASK 1

See Fig. 10.

## B.2  EVALUATE THE DIVERSITY OF TASK 1

To further quantify behavioral diversity in our datasets, we evaluated three distinct sources: the human-human dataset from Moving Out Task 1, a dataset collected from four human experts, and trajectories generated by a trained MAPPO agent. We report diversity across both trajectory- and state-level dimensions. At the trajectory level, we use Dynamic Time Warping (DTW) to compute pairwise distances between all trajectories within a dataset; higher mean and variance indicate greater dissimilarity in path shapes. At the state level, we measure spatial coverage using two complementary

Visualization of Collected Trajectories in Task 1

Figure 10: Visualization of data collected in Task 1 from four groups (each with two players) and from the complete dataset. Blue dots denote the positions of the blue agent and red dots denote the positions of the red agent. The visualizations show clear differences across groups. At the aggregated level, the dataset captures both human behavioral preferences (e.g., preferred paths and object-grasping locations) and broad state coverage.

metrics: Kernel Density Estimation (KDE) entropy over agent positions and an additional coverage distance metric inspired by (Fu et al., 2023), which computes the average pairwise distance between trajectories using an RBF kernel. Higher values for both metrics indicate broader exploration of the map. As shown in Table 1, the Task 1 dataset consistently achieves higher scores across all metrics, confirming that data aggregated from 36 human players exhibits substantially greater behavioral diversity compared to expert demonstrations and RL-generated trajectories. This diversity provides a rich foundation for training adaptive collaboration policies and benchmarking generalization.

## C    COMPARISON WITH ORACLE SIMULATION

| | NFD↑ | | Prediction Accuracy | |
|---|---|---|---|---|
| | Task 1 | Task 2 | Task 1 | Task 2 |
| DP + BASS | 0.5733 | 0.5535 | 0.6250 | 0.4870 |
| DP + BASS w/Oracle Simulator | 0.5875 | 0.6209 | N/A | N/A |

Table 4: Performance of different simulation strategies. The oracle simulator serves as the upper bound for our method.

We compare the task completion (NFD) and prediction accuracy of actions against the oracle simulator (i.e., the 2D physics engine) in Table 4. We compute the prediction accuracy by comparing the actions selected using our next state predictor versus the actions selected using the oracle simulator. The oracle simulator serves as the upper bound for our action selection method since it provides the ground-truth next states. We observe that our model achieves higher accuracy in Task 1, with results that are closer to those of the oracle simulator. This is because Task 1 uses a fixed map, while Task 2 trains on randomized states.

### C.1    ABLATION STUDY

**Ablations** Table 5 shows the ablation of each component. Adding augmentation and simulation components improves task completion TCR and NFD compared to their base models. When using all components (full BASS), they achieve the highest overall performance in most cases.

|                          | Task 1 |       | Task 2 |       |
| ------------------------ | ------ | ----- | ------ | ----- |
| Methods                  | TCR↑   | NFD↑  | TCR↑   | NFD↑  |
| GRU                      | 0.3070 | 0.3674 | 0.2582 | 0.3935 |
| + BASS w/o Simulation    | 0.4117 | 0.4396 | 0.3333 | 0.4141 |
| + BASS w/o Augmentation  | 0.3531 | 0.4047 | **0.3670** | 0.4246 |
| + Full BASS              | **0.4120** | **0.4454** | 0.3414 | **0.4410** |
| Diffusion Policy (DP)    | 0.3829 | 0.4818 | 0.3125 | 0.4526 |
| + BASS w/o Simulation    | 0.4028 | 0.5114 | 0.3569 | 0.4908 |
| + BASS w/o Augmentation  | 0.4741 | 0.5561 | 0.4200 | 0.5187 |
| + Full BASS              | **0.5027** | **0.5707** | **0.4348** | **0.5535** |

Table 5: Ablations showing the impact of each component, we show BASS with GRU and DP backbones.

## D  ROLLOUT EXAMPLE

Figure 11 shows an example rollout on Task 2. We only present one example here; additional maps and tasks can be found in the supplementary video.

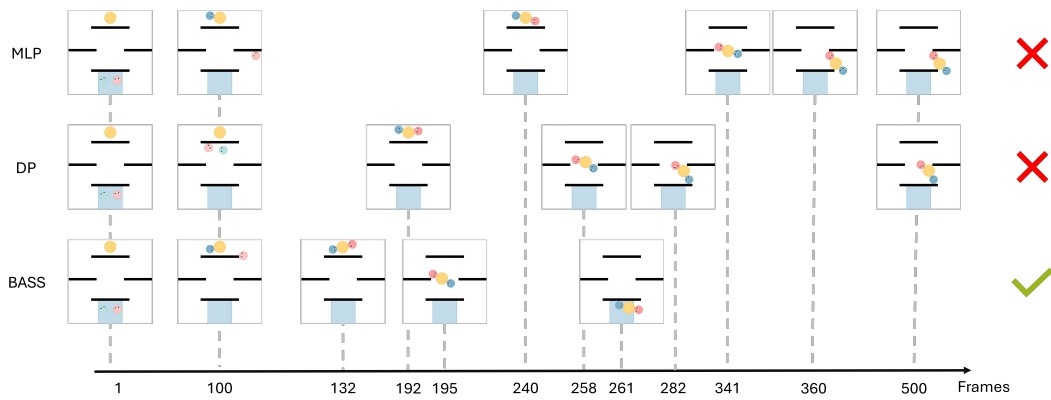

Figure 11: Comparison of rollouts on Task 2 between our method (BASS), MLP, and DP. The horizontal axis denotes frames as a proxy for time. Our method successfully completes the task at frame 261, whereas both MLP and DP get stuck at an intersection. This highlights the challenge of Task 2, where handling the large fixed-mass circular object is particularly difficult.

## E  MAPPO TRAINING SETTING

To train MAPPO, we integrate the Moving Out environment into the BenchMARL (Bettini et al., 2024) multi-agent RL library. Our approach to MAPPO training was designed to align with the objectives of Task 1 and Task 2, which were initially conceptualized with dataset-driven methods in mind. We adapted the conditions for MAPPO as follows:

For Task 1, which originally involved training on data collected from some human players and testing on data from unseen human players, we interpret this as a zero-shot coordination challenge for MAPPO. This setup evaluates their ability to develop coordination strategies from scratch in the absence of direct human examples.

For Task 2, the initial idea was to train on maps with diverse physical characteristics and then evaluate generalization to environments with unseen physical features. To mirror this for MAPPO, the agents are trained on maps where various physical properties (object masses, shapes, and sizes) are randomized, similar to the randomization process used during data collection for behavior cloning. Following this training phase, MAPPO's performance is then evaluated on maps with fixed physical characteristics that were not encountered during training.

### E.1  HYPERPARAMETERS

Table 6: Summary of Parameters for MAPPO

| Parameter Name | Value |
|---|---|
| Share Policy Parameters | True |
| Share Policy Critic | True |
| Gamma ($\gamma$) | 0.99 |
| Learning Rate | 0.00005 |
| Adam Epsilon | 0.000001 |
| Clip Gradient Norm | True |
| Clip Gradient Value | 5 |
| Soft Target Update | True |
| Polyak Tau ($\tau$) | 0.005 |
| Hard Target Update Frequency | 5 |
| Initial Exploration Epsilon | 0.8 |
| Final Exploration Epsilon | 0.01 |
| Clip Epsilon | 0.2 |
| Critic Coefficient | 1.0 |
| Critic Loss Type | l2 |
| Entropy Coefficient | 0 |
| Lambda ($\lambda$) for GAE | 0.9 |
| Max Cycles Per Episode | 1000 |
| Max Frames | 30,000,000 |
| On-Policy Collected Frames Per Batch | 6000 |
| On-Policy Environments Per Worker | 10 |
| On-Policy Minibatch Iterations | 45 |
| On-Policy Minibatch Size | 400 |
| Model Type | MLP |
| Linear Layer Sizes | [256, 256] |
| Activation Function | torch.nn.Tanh |

For coordination maps, due to the greater distance from the initial explorer positions to the target items and the presence of more walls, we increased max_cycles_per_episode from 1000 to 3000. Concurrently, we adjusted entropy_coef to 0.00065 and gamma to 0.92 for these maps.

## F    REWARD SETTING

### F.1    DENSE REWARD SETTING

The dense reward is based on the change in distance $\Delta d = d_{\text{prev}} - d_{\text{curr}}$, scaled by a factor $\gamma = 20$, where $d_{\text{prev}}$ and $d_{\text{curr}}$ denote the agent's distance to the current target at the previous and current timestep, respectively. When the agent is not holding an object, the target is either the nearest unheld item or a middle/large item currently being moved by another agent that requires assistance. When the agent is holding an object, the target becomes the goal region. At each timestep, the agent receives a reward of $\Delta d \times \gamma$. See Tab. 7 for more details.

Additionally, there are special rewards tailored for specific maps. In Map 11 (Four Corners), for instance, two agents need to hold the two short sides of a rectangular item to more easily pass through a path successfully. Therefore, to encourage this, the reward calculation for the agents' distance to this item has been modified: instead of being based on the distance to the item's center point, it is now calculated based on the distance to its two short sides. This change is designed to encourage the agents to grasp the rectangle by its short ends.

### F.2    DOES MAPPO WORK IN MOVING OUT WITH SPARSE REWARD SETTING?

The primary challenge in Moving Out lies in its significantly larger and more complex state space. Within such an expansive environment, agents who take random exploration struggle to successfully

Table 7: Dense Reward Settings

| Primary State / Event | Specific Condition | Reward Value |
|---|---|---|
| *A. Distance-based Rewards* | | |
| Agent not holding an item | Agent moves closer to the nearest available item | $\Delta d \times \gamma$ |
| | Agent moves closer to a middle or large item currently held by another agent | $\Delta d \times \gamma$ |
| Agent holding an item | Agent moves closer to the nearest goal region | $\Delta d \times \gamma$ |
| *B. Event-based Rewards: Agent Holds an Item* | | |
| Agent successfully holds an item | Default reward for picking up | $+0.5$ |
| | *Exception:* If another agent is already holding other middle or large item at this time | $-0.5$ (total for this hold event) |
| | *Exception:* If the item picked up was already located within a goal region | $-0.5$ (total for this hold event) |
| *C. Event-based Rewards: Agent Unholds an Item* | | |
| Agent successfully unholds an item | Item is released inside a goal region | $+0.5$ |
| | Item is released *not* inside a goal region | $-0.5$ |
| | *Exception:* If another agent needs help, holding a large or middle item outside the goal region, at the moment of unholding. | $+0.5$ (additive) |
| *D. Time-based Reward (Step Cost)* | | |
| Each timestep | Agent exists in the environment | $-0.01$ |

complete the multi-step tasks required to reach goal states and thus rarely receive the sparse or event-based rewards crucial for learning. Consequently, sparse reward formulations currently appear insufficient for effective policy learning in Moving Out.

MAPPO algorithms employing sparse or event-based rewards have achieved notable success in environments such as Overcooked-AI. This success can be largely attributed to the characteristics of Overcooked-AI, specifically its discrete action-state space and relatively compact overall state space. These features allow agents to encounter rewarding events with sufficient frequency through exploration, even when rewards are not dense, facilitating effective policy learning.

In Overcooked, the state-action space is small and discrete, with only tens of possible states and six possible actions, effectively rendering it a tabular setting. In contrast, our environment features continuous state and action spaces, states include precise map coordinates, and actions involve continuous control over speed and direction. Although RL is relatively easy for small discrete space, extending methods to handle continuous space is non-trivial.

Moreover, the tasks in Overcooked are relatively simple: agents fetch onions from a fixed area and deliver them using plates. Onions and plates are homogeneous, unlimited, and confined to designated regions. Once picked up, items can only be placed in predefined locations for handoff, simplifying coordination between agents.

By comparison, our tasks are significantly more complex with additional physical constraints. First, the items in our environment are heterogeneous, which are randomized in shape, size, and initial position. Thus, agents must learn to generalize over combinations of all possible scenarios. Second, unlike Overcooked, where items can only be placed in fixed zones, our agents can place items

anywhere on the map. This greatly increases the difficulty of learning how to transfer items to target locations or hand them off between agents, especially in a continuous space. Additionally, our framework requires agents to engage in a wider range of collaborative behaviors beyond simple item passing—for instance, jointly moving large objects or coordinating to rotate items in tight spaces like wall corners. This diversity of collaboration types introduces further complexity.

## G  COMPARATIVE ANALYSIS OF THE BEHAVIORS OF BC AND RL AGENTS

The fundamental difference between Behavior Cloning (BC) and MAPPO lies in their learning mechanisms and resulting agent behaviors. BC methods are inherently data-driven, leading to policies whose actions and overall effectiveness closely mirror the human behaviors captured in the training dataset. In contrast, MAPPO, as a reinforcement learning (RL) approach, develops behaviors that are strongly guided by the specific design of its dense reward function.

This distinction is evident in specific scenarios. For instance, on Map 6 (Distance Priority), both agents have their closest middle-sized items. However, human demonstration data frequently shows a strategy of first securing two smaller items before returning to move a middle-sized item together. A MAPPO agent, guided by a dense reward that incentivizes moving the nearest object, will typically prioritize the closer middle-sized item. If two such items are equidistant to respective agents (e.g., a pink agent targeting a yellow star and a blue agent targeting a blue circle), the initial actions will be independent. The coordination emerges when one agent successfully grasps a middle-sized item; the reward structure then incentivizes the other agent to assist with that specific item. Thus, the RL behavior can appear as a race to secure a primary middle-sized object, with the "loser" then being redirected by rewards to help the "winner." BC models on Map 6 (Distance Priority), however, reflect the diversity of the human dataset. This dataset contains instances of both "small-items-first" and "middle-item-first" strategies. Consequently, a BC agent might exhibit behaviors where one agent targets a middle-sized item while the other simultaneously attempts to move a small item, reflecting a momentary misalignment as different agents emulate distinct strategies observed in the human data.

Map 11 (Four Corners) further illustrates these differences. Here, two agents might each have two items at an equal distance, making multiple initial moves potentially optimal. In our MAPPO training, agents often exhibit initial movements that appear somewhat exploratory or randomized until one agent commits to and grasps a large item. At this point, the dense reward system effectively directs the other agent to provide assistance. Conversely, BC models on Map 11 (Four Corners) tend to display more decisive and rapidly aligned behavior from the start. Observations of the human dataset for this map revealed a common leader-follower dynamic, where one player (e.g., the blue agent) consistently follows the lead of the other (e.g., the pink agent). If the pink agent, for example, decisively moves towards an upper pink square, the blue agent often follows suit immediately to assist. As a result, BC models rarely exhibit prolonged periods of uncoordinated or hesitant movement before aligning on a common goal.

In summary, BC methods excel at reproducing observed human behaviors, including their specific strategies and inherent diversity. RL approaches like MAPPO, while capable of discovering effective strategies, are highly sensitive to the nuances of reward function design. Even slight modifications to the reward signals can lead to significant and sometimes qualitatively different emergent behaviors in the trained agents.

## H  ADDITIONAL VALIDATION OF BEHAVIOR AUGMENTATION

**Behavior mismatch in sub-trajectory recombination.** Our recombination strategy is explicitly designed to avoid the type of inconsistency that was described. Specifically, we only perform sub-trajectory swapping when the fixed agent (e.g., agent A) has the same start and end poses across two trajectories. This ensures that agent A is pursuing the same local goal in both cases, regardless of the specific behavior of the partner.

For example, suppose in trajectory $\tau_1$, agent A performs action sequence $a_1$ while agent B performs $b_1$, and in $\tau_2$, A performs $a_2$ while B performs $b_2$. If $a_1$ and $a_2$ share the same start and end states, we can create two new combinations: $(a_1, b_2)$ and $(a_2, b_1)$. These are valid because both $b_1$ and $b_2$ were originally compatible with different variants of A's strategy toward the same goal. As such,

swapping B's behavior does not interfere with A's intent. This preserves behavioral diversity while ensuring trajectory-level coherence.

**Additional validation.** To ensure consistency, we identify sub-trajectories with matching start and end poses, so that the recombined agent behaviors maintain the same intention and goal. We have validated this approach with over 99% success rate in producing physically valid trajectories. In addition, we also confirm that the diversity of behavior increases after augmentation (Entropy improves from 0.88 to 0.95).

# I  DETAILS OF EVALUATION METRICS

To assess human-AI collaboration in *Moving Out*, we design metrics that go beyond final task success to capture the *quality of physical collaboration*. While prior works such as Overcooked-AI mainly rely on task completion, this is insufficient in physically grounded settings, where interactions involve continuous control, object dynamics, and force alignment. We therefore complement **Task Completion Rate (TCR)** with three additional metrics—**Normalized Final Distance (NFD)**, **Waiting Time (WT)**, and **Action Consistency (AC)**—each targeting a different aspect of collaboration under physical constraints.

**Task Completion Rate (TCR).** TCR measures the proportion of objects successfully delivered to goal regions, weighted by size:

$$TCR = \frac{\sum w_i \mathbb{I}(o_i \text{ delivered})}{\sum w_i},$$

where $w_i = 1$ (small) or 2 (middle/large). Range: [0,1]. TCR captures the final outcome of collaboration, but by itself cannot distinguish between failed attempts with meaningful progress and those with no progress.

**Normalized Final Distance (NFD).** NFD quantifies partial progress by measuring the reduction in object-goal distance:

$$NFD = 1 - \frac{\sum_{i=1}^{N} d_i^{\text{final}}}{\sum_{i=1}^{N} d_i^{\text{initial}}},$$

where $d_i^{\text{initial}}$ and $d_i^{\text{final}}$ are the object's initial and final distances to the target. This is critical in physical environments where objects may get stuck due to collisions or narrow passages. A case with high NFD but low TCR indicates that agents made progress but failed to overcome physical constraints.

**Waiting Time (WT).** WT captures how agents coordinate when joint effort is required:

$$WT = \sum_{t \in \mathcal{W}} (t_{\text{end}}^t - t_{\text{start}}^t),$$

where $\mathcal{W}$ is the set of intervals when an agent holds a middle/large object but must wait for help. High WT may reflect either poor recognition of the need for help or inefficiency in navigating physical obstacles. Thus, it measures awareness and responsiveness in physically grounded collaboration.

**Action Consistency (AC).** As illustrated in Fig. 12, AC measures how well two agents align their applied forces during joint manipulation:

$$AC = \frac{1}{T} \sum_{t=0}^{T-1} \frac{\|(\vec{f}_1^t + \vec{f}_2^t) \cdot \vec{d}_t\|}{\|\vec{f}_1^t\| + \|\vec{f}_2^t\|},$$

where $\vec{f}_1^t, \vec{f}_2^t$ are the forces applied at time $t$, $\vec{d}_t$ is the unit vector connecting agent positions, and $T$ is the number of timesteps. This metric captures coordination quality: agents are most effective when their forces are aligned, and receive low scores when their efforts cancel out (e.g., one pushing forward, the other pulling back).

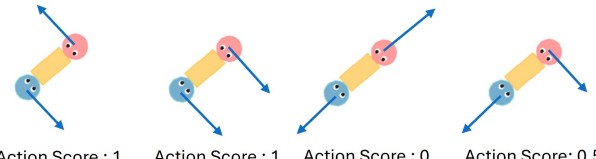

Action Score : 1    Action Score : 1    Action Score : 0    Action Score: 0.5

Figure 12: Example of action consistency (AC) calculation. Effective collaborative work receives high scores, while opposing forces canceling each other lead to low scores.

Together, these four metrics provide a comprehensive evaluation of human-AI collaboration in physically grounded tasks: TCR reflects task success, NFD measures partial progress under constraints, WT captures coordination in joint effort, and AC quantifies the efficiency of physical interaction. This combination moves beyond symbolic settings and offers a richer view of how collaboration unfolds in continuous, low-level environments.

## J    IMPLEMENTATION DETAILS

### J.1    ENVIRONMENT DETAILS

#### J.1.1    OBSERVATION ENCODING

**State Observation**    Our observation encoding is ego-centric and represents all information as a one-dimensional vector. The encoded information includes:

- Self: Position and angle, with angles $\theta$ represented using $[\cos\theta, \sin\theta]$. A boolean value indicates whether the agent is holding an item (True/False).
- Partner: Position, angle, and whether it is holding.
- Items: Each item is encoded with position, angle, size, category, and shape. Category and shape use a one-hot encoding.

When training on a single map, the walls and goal region remain unchanged, so we do not encode them. However, when training across different maps, we include their encoding:

- Walls: Represented by the $(x, y)$ coordinates of the top-left and bottom-right corners.
- Goal Region: Represented the save as walls. The top-left and bottom-right corners.

#### J.1.2    ACTION ENCODING

The agent's action space has four values:

- The movement distance (forward or backward).
- The target angle (encoded using $\cos$ and $\sin$).
- The grasping action: 1 means grasp or release, 0 means no change.

### J.2    BASELINE DETAILS

- **Diffusion Policy**: We follow the original implementation by (Chi et al., 2023) for the model architecture, which employs a 1D U-Net to generate action sequences. The observation, prediction, and executable horizons are set to 2, 8, and 4, respectively. Training is performed using the Adam optimizer with 1k epochs, 1024 batch size, and 0.001 learning rate. The diffusion steps are 36. The grasp action is encoded by one-hot encoding.
- **MLP** The MLP model consists of 3 fully connected layers with Tanh activation and hidden_dim 2048. It concatenates one past state and one current state as input and predicts actions for the next 8 steps. Training is performed using the Adam optimizer with 1k epochs, 1024 batch size, and 0.001 learning rate. It optimizes a combination of mean squared error (MSE) loss for movement outputs and cross-entropy loss for grasp action predictions.

- **GRU** uses a GRU layer followed by 3 fully connected layers with Tanh activation and hidden_dim 2048. It takes one past state and the current state as input and predicts actions for the next 4 steps. The model processes sequential data and learns action patterns based on previous movements. Training is performed using the Adam optimizer with 1k epochs, 1024 batch size, and 0.001 learning rate. It optimizes a combination of mean squared error (MSE) loss for movement outputs and cross-entropy loss for grasp action predictions.

## J.3   BASS DETAILS

- **Dynamics Model** The Autoencoder consists of an encoder and a decoder, both made of two linear layers. They use ReLU as the activation function, and each layer has 128 units. The latent space has 32 dimensions. The dynamics Model is a two-layer MLP (Multi-Layer Perceptron). Each hidden layer has 128 units. During training, the two autoencoders and the dynamic model are trained together. Additionally, we also explored fine-tuning the second AE from the first. Our ablation on selected Maps 2, 6, & 9 shows the following average NFDs: 1) Joint training: 0.55, 2) Fine-tuning the second AE from the first AE: 0.50, 3) Training two AEs separately: 0.48.

- **Partner Action Predictor** The Partner Action Predictor can be designed based on the application. In some cases, it can be the same as the action policy, but with a small change: it swaps the agent's state with the partner's state. This allows the model to predict the partner's action from their perspective.

**Behavior Augmentation and Recombination Sub-Trajectories**   In behavior augmentation, we add noise with a mean of 0 and a standard deviation of 0.002. In recombination sub-trajectories, since two points in a continuous space are almost never the same, we set a tolerance value. We discretize the environment into a $48 \times 48$ grid. If the robot's start and end points are in the same grid cell, we treat them as the same point.

**Normalized Final Distance Calculation**   Many maps have walls, so we cannot use Euclidean distance. To improve efficiency, we discretize the environment into a $48 \times 48$ grid. We use the BFS algorithm to compute the distance from the item to the Goal Region.

**Can BASS be used as a standalone Method beyond Moving Out?**   BASS is a standalone method composed of two components. Together, they make BASS applicable across various behavior cloning methods outside of Moving Out, as discussed. We tested BASS on a widely used human-AI collaboration environment, Overcooked AI, specifically, the "Cramped Room" map. The results showed that DP+BASS improved the score by  15% compared to DP alone. This demonstrates that BASS is not limited to Moving Out.

# K   STUDY ON ACTION SAMPLING TIMES

We conducted an ablation on the number and strategy of action samples used in BASS. In our implementation, each candidate action is generated by independently sampling from the policy and partner predictor up to four times, producing four simulation rollouts. This setting provides a good balance between performance and efficiency, supporting real-time human evaluation at 10Hz.

To compare alternatives, we tested three strategies: 1) Independent 4× sampling (current setting); 2) 2×2 combination (two samples from each agent, combined into four rollouts); 3) 4×4 combination (four samples from each, combined into sixteen rollouts).

Results are summarized in Table 8. The 2×2 strategy, despite using the same number of simulations as the independent setting, consistently underperforms. Independent sampling has higher chance to capture critical joint transitions, e.g., resolving a stuck state. The 4×4 combination achieves the best accuracy, but requires 16 rollouts and increases inference time from 69 ms to 210 ms, which disrupts real-time human evaluation at 10Hz.

We therefore adopt the independent 4× sampling scheme in BASS, as it balances accuracy with the real-time feasibility required for human-in-the-loop collaboration.

| | Task 1 | | | | Task 2 | | | |
|---|---|---|---|---|---|---|---|---|
| | TCR↑ | NFD↑ | WT↓ | AC↑ | TCR↑ | NFD↑ | WT↓ | AC↑ |
| Sample 4× independently | 0.5027 | 0.5707 | 0.3448 | 0.8615 | 0.4348 | 0.5535 | 0.3096 | 0.8474 |
| 2×2 combination | 0.4522 | 0.5261 | 0.3485 | 0.8462 | 0.4158 | 0.5137 | 0.3101 | 0.8460 |
| 4×4 combination | **0.5161** | **0.5847** | **0.3390** | **0.8727** | **0.4396** | **0.5638** | **0.3087** | **0.8559** |

Table 8: Results of sampling and combination strategies.

## L  BASS ANALYSIS

### L.1  BASS MODULE ANALYSIS

#### L.1.1  NEXT-STATE PREDICTION ACCURACY

We evaluate the accuracy of the next-state prediction module by computing the L2 distance between the predicted state and the ground-truth state from the oracle simulator. Two baselines are included: a GRU-based predictor and a random state generator.

| L2 Norm (↓) | Task 1 | Task 2 |
|---|---|---|
| BASS | **0.0010** | **0.0028** |
| GRU | 0.0196 | 0.0331 |
| Random States | 2.7576 | 3.2594 |

Table 9: Next-state prediction accuracy across tasks. Lower is better.

These results show that BASS more accurately captures physical dynamics compared to both the GRU baseline and random guessing, supporting the effectiveness of the learned dynamics model in simulation and action selection.

#### L.1.2  MODELING DIVERSE HUMAN BEHAVIORS

To handle diverse human behaviors, our approach models partner actions as a conditional distribution learned from demonstrations. The latent dynamics model captures this diversity by representing multiple likely behaviors under the same state, instead of committing to a single mode.

We use a Diffusion Policy, which effectively models multimodal action distributions by sampling from different noise inputs (Li et al., 2024b; Chi et al., 2023). This enables the model to generate different possible partner responses, providing probabilistic reasoning that aligns with human intuition.

#### L.1.3  PARTNER ACTION PREDICTION ACCURACY

Since the action space is continuous, prediction accuracy is evaluated using the relative error between predicted and ground-truth actions. With a 10% error tolerance, the predictor achieves an accuracy of 71.45%; relaxing the tolerance to 20% increases accuracy to 90.24%. These results indicate that the partner action predictor provides sufficiently accurate estimates to support effective next-state prediction and action selection within our framework.

#### L.1.4  EFFECT OF RANDOM PARTNER ACTIONS

To assess the importance of accurate partner action prediction, we conduct an ablation where the partner's actions are randomly sampled during the simulation step. Since our action selection process considers four candidates, random guessing introduces uncertainty and degrades the ability to make correct selections between predicted future states.

#### L.1.5  COMPARISON WITH ALTERNATIVE NEXT-STATE PREDICTION MODELS

To evaluate the effectiveness of our proposed dynamics model, we compare it with two alternatives: a GRU-based predictor and a qVAE model for next-state prediction. As shown in Table 11, our

| | TCR (↑) | | Action Selection Accuracy (↑) | |
|---|---|---|---|---|
| | Task 1 | Task 2 | Task 1 | Task 2 |
| BASS (Ours) | 0.5027 | 0.4348 | 0.6250 | 0.4870 |
| BASS (Random partner action) | 0.3966 | 0.3650 | 0.2542 | 0.2484 |
| BASS w/ Oracle Simulator | 0.5421 | 0.5334 | 1.0000 | 1.0000 |

Table 10: Impact of replacing the learned partner model with random actions. Both task performance (TCR) and action selection accuracy drop significantly.

model significantly outperforms both in prediction accuracy and final task performance (NFD). The GRU baseline performs close to random guessing, indicating difficulty in learning accurate transition dynamics. The qVAE model performs slightly better, but still struggles, likely due to the large continuous state space, where discretized latent codes are insufficient to represent fine-grained physical interactions. These results highlight the importance of our autoencoder-based latent dynamics model in capturing physical transitions effectively.

| | NFD (↑) | | Prediction Accuracy (↑) | |
|---|---|---|---|---|
| | Task 1 | Task 2 | Task 1 | Task 2 |
| BASS (Ours) | 0.5733 | 0.5535 | 0.6250 | 0.4870 |
| BASS (GRU) | 0.5048 | 0.4965 | 0.2487 | 0.2598 |
| BASS (qVAE) | 0.5113 | 0.5032 | 0.3102 | 0.2722 |
| BASS w/ Oracle Simulator | 0.5875 | 0.6209 | N/A | N/A |

Table 11: Comparison of next-state prediction models. Our dynamics model outperforms GRU and qVAE in both prediction accuracy and task performance.

## L.2 Comparison with Single Agent BASS

Our method extends augmentation, simulation, and action selection to a multi-agent collaborative setting, which introduces challenges fundamentally different from single-agent environments. In collaborative manipulation, both agents jointly affect the shared object, so a valid sub-trajectory must be compatible with the partner's motion. By contrast, single-agent recombination only checks whether a segment starts or ends from a similar state, without ensuring that the segment represents the same behavioral goal. As formally defined in the Appendix, the single-agent version treats two segments as compatible if either $s_{t_1}^i \approx \hat{s}_{t_3}^i$ or $s_{t_2}^i \approx \hat{s}_{t_4}^i$. Because this criterion ignores the evolution of the partner's motion and does not ensure that the two segments correspond to the same part of the task, the recombined trajectories can easily violate the coordinated patterns required for joint manipulation.

Our multi-agent recombination module avoids this issue by requiring that agent $i$ begins and ends the segment in nearly the same physical situation in both demonstrations. This ensures that agent $i$ is performing the same portion of the task, so the partner's subsequence can be safely swapped while keeping agent $i$'s behavior consistent throughout the segment. As shown in Table 12, the single-agent recombination baseline increases diversity metrics but degrades collaboration performance, indicating that the generated trajectories no longer reflect valid joint behaviors. In contrast, our multi-agent recombination increases both diversity and cooperative performance, demonstrating that preserving cross-agent compatibility is essential for effective augmentation in collaborative settings.

We further examine the action simulation and selection module. If the next-state prediction model considers only one agent's future action while ignoring the partner's state evolution, its effectiveness is greatly diminished. As shown in Table 13, a single-agent simulation model yields performance close to disabling simulation entirely. This highlights the importance of modeling the coupled dynamics of both agents and further confirms that the proposed approach cannot be reduced to a combination of single-agent components.

Table 12: Comparison between multi-agent recombination and single-agent recombination on Moving Out Task 1. Multi-agent recombination increases diversity while preserving coordinated behavior, whereas single-agent recombination increases diversity but harms collaboration performance.

| Method | Diversity | | | | Performance | | | |
|---|---|---|---|---|---|---|---|---|
| | DTW Mean | DTW Var | Entropy (KDE) | Coverage (RBF) | TCR($\uparrow$) | NFD($\uparrow$) | WT($\downarrow$) | AC($\uparrow$) |
| Multi-agent Recombination (Ours) | 7.526 | 6.612 | 0.892 | 0.910 | **0.403** | **0.511** | **0.308** | **0.840** |
| Single-agent Recombination | **7.741** | **6.722** | **0.897** | **0.919** | 0.368 | 0.451 | 0.338 | 0.839 |
| No Recombination | 7.013 | 6.065 | 0.888 | 0.899 | 0.383 | 0.482 | 0.345 | 0.824 |

Table 13: Comparison of multi-agent vs. single-agent action simulation on Moving Out Task 2. Without considering the partner's future state, the benefit of simulation is greatly reduced.

| Method | TCR($\uparrow$) | NFD($\uparrow$) | WT($\downarrow$) | AC($\uparrow$) |
|---|---|---|---|---|
| Multi-agent Action Simulation (Ours) | **0.420** | **0.554** | **0.310** | **0.848** |
| Single-agent Action Simulation | 0.319 | 0.458 | 0.327 | 0.833 |
| No Action Simulation | 0.313 | 0.452 | 0.335 | 0.821 |

Overall, these results demonstrate that our method must explicitly account for multi-agent coordination constraints. The approach is not a direct extension of single-agent techniques but instead relies on mechanisms specifically designed to maintain cross-agent alignment. While formal theoretical guarantees are beyond the scope of this work, the empirical evidence highlights the importance of multi-agent structure, and we plan to investigate deeper theoretical characterizations in future work.

## M EVALUATION PROTOCOL FOR TASK 1

**Human-based evaluation.** Since Task 1 is designed to evaluate adaptation to unseen human behaviors, our primary setup requires agents to play with new human participants. While this is the most direct evaluation of human-AI collaboration, it raises concerns about reproducibility because new participants are required for each run.

**Cross-group reproducible protocol.** To address this, we design a reproducible protocol inspired by cross-model evaluation. We randomly split the human demonstrations into two groups, each containing data from different participants. Two models are trained separately on each group and then evaluated by playing with each other. This setup emulates collaboration with unseen partner behavior while remaining fully reproducible.

**Results.** Table 14 compares the performance of Diffusion Policy (DP) and our method (BASS) when paired with models trained on the same group vs. a different group. We also report the percentage drop (or increase) in performance when moving from same-group to cross-group evaluation. Results show that models perform better when paired with a model trained on the same group (more aligned behavior), but BASS consistently outperforms DP when paired with unseen human behaviors, demonstrating stronger generalization.

| Setting | Method | TCR ($\uparrow$) | NFD ($\uparrow$) | WT ($\downarrow$) | AC ($\uparrow$) |
|---|---|---|---|---|---|
| Same-Group | DP | 0.3233 | 0.5367 | 0.3789 | 0.8163 |
| | DP/BASS | 0.3503 | 0.5724 | 0.3598 | 0.8337 |
| Cross-Group | DP | 0.2563 (-20.72%) | 0.4589 (-14.50%) | 0.4249 (+12.15%) | 0.7854 (-3.78%) |
| | DP/BASS | 0.3010 (-14.07%) | 0.5197 (-9.22%) | 0.3899 (+8.37%) | 0.8099 (-2.86%) |

Table 14: Cross-group evaluation protocol for Task 1. Performance drops (%) are measured relative to same-group evaluation.

## N  WHY CHOOSE NORMALIZED FINAL DISTANCE IN BASS?

We chose Normalized Final Distance (NFD) because it directly reflects task progress in physically grounded collaboration. When two agents move an object together, actions that successfully reduce the distance between objects and the goal region indicate effective cooperation, even when navigating around obstacles like walls. Thus, maximizing NFD considers both physical feasibility and cooperation efficiency.

We also experimented with a multi-objective scoring using both NFD and Action Consistency (AC), to encourage not only progress but also force alignment. The trade-off is shown below:

| Task 1 | NFD ($\uparrow$) | AC ($\uparrow$) |
| --- | --- | --- |
| BASS with NFD | 0.5733 | 0.8615 |
| BASS with NFD+AC | 0.5683 | 0.9127 |

Table 15: Comparison of using NFD vs. NFD+AC as objectives.

Although this combined objective improved AC, we found that NFD dropped slightly. We chose to prioritize NFD in the paper for its ability to capture physical task progress.

## O  BEHAVIOR AUGMENTATION DETAILS

Our augmentation strategy involves two techniques:

**Generating New States by Perturbing the Partner's Pose** For a given trajectory, we generate new states by introducing noise to the partner's pose while keeping all other state variables unchanged. This perturbation creates additional observation variations in training data, allowing the agent to experience a broader range of possible partner behaviors. Since human actions naturally vary, this approach helps improve the agent's robustness to small deviations in the partner's movements while maintaining its own task objectives. This perturbation is expressed as $\tilde{p}_{\text{partner}} = p_{\text{partner}} + \epsilon, \epsilon \sim \mathcal{N}(0, \sigma^2)$ where $p_{\text{partner}}$ is the original partner's pose, $\epsilon$ is Gaussian noise with mean $0$ and variance $\sigma^2$, and $\tilde{p}_{\text{partner}}$ is the perturbed pose used to generate new state variations.

**Recombination of Sub-Trajectories** Each global state $s_t$ can be decomposed into $s_t = \left(s_t^i, \ s_t^j, \ s_t^e\right)$, where $s_t^i$ and $s_t^j$ are the individual states of agent $i$ and $j$, and $s_t^e$ captures the remaining environment-specific information. Given a trajectory $\tau = \{(s_t, a_t)\}_{t=1:T}$, we extract three sequences: $\tau^i = \left\{(s_t^i, a_t^i)\right\}_{t=1:T}$ is the state-action sequence of agent $i$; similarly $\tau^j$ is the state-action sequence of agent $j$ and $\tau_e$ is the sequence of environment information. We have $\tau = \tau^i \cup \tau^j \cup \tau^e$. Moreover, let $\tau_t^i = (s_t^i, a_t^i)$ be the $t$-th state-action pair of agent $i$, and define $\tau_{t_1:t_2}^i = (s_{t_1}^i, a_{t_1}^i, \cdots, s_{t_2}^i, a_{t_2}^i)$ as the continuous sub-trajectory of $\tau^i$ from $t_1$ to $t_2$. We can define $i$'s trajectory composed of sub-trajectories $\tau^i = \tau_{1:t_1-1}^i \cup \tau_{t_1:t_2}^i \cup \tau_{t_2+1:T}^i$; and similarly for $j$.

Given $\tau$ and two time step $t_1, t_2$, we can search for another trajectory $\hat{\tau}$ in the dataset such that $\hat{\tau}_{t_1}^i = \tau_{t_1}^i$ and $\hat{\tau}_{t_2}^i = \tau_{t_2}^i$. We can then construct two new trajectories by swapping agent $j$'s subsequences between $t_1$ and $t_2$:

$$\tau^i \cup \left(\tau_{1:t_1-1}^j \cup \hat{\tau}_{t_1:t_2}^j \cup \tau_{t_2+1:T}^j\right) \cup \tau^e \quad \text{and} \quad \hat{\tau}^i \cup \left(\hat{\tau}_{1:t_1-1}^j \cup \tau_{t_1:t_2}^j \cup \hat{\tau}_{t_2+1:T}^j\right) \cup \hat{\tau}^e$$

By aligning the start and end of agent $i$'s sub-trajectory, the generated trajectories maintain temporal consistency for agent $i$ while introducing a different sequence. This approach enriches the training set with new, valid trajectories where agent $i$'s behavior is fixed and the partner's varies.

### O.1  VISUALIZATION EXAMPLE OF RECOMBINATION

Fig. 13 illustrates an example of our trajectory recombination process. Consider two human demonstrations (or two trajectories in the dataset). If a trajectory segment from the red agent in both demonstrations starts from nearly the same initial state, we treat these segments as behaviorally compatible. This implies that, under this specific behavior of the red agent, the corresponding

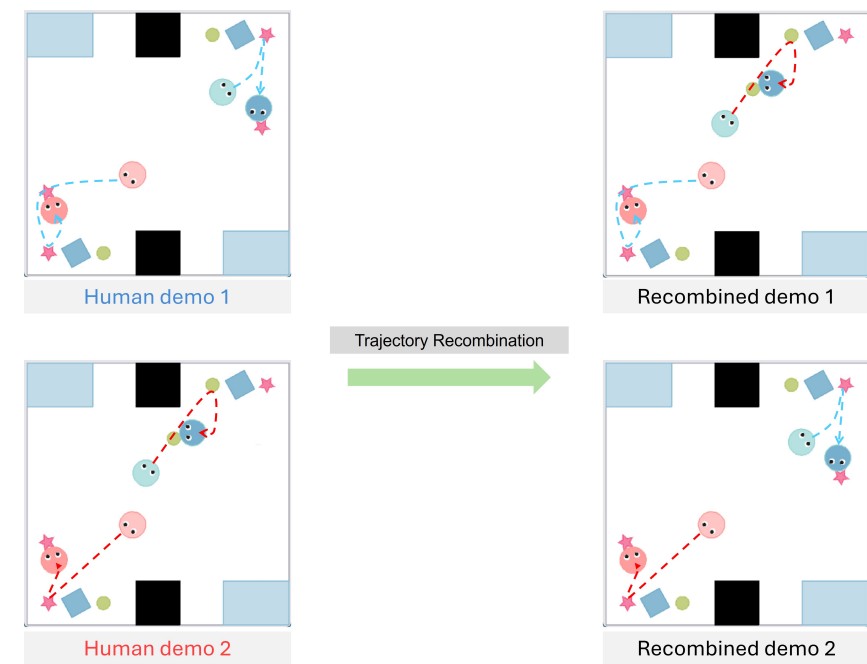

Figure 13: Two human demonstrations (left) contain trajectory segments where the red agent begins from almost identical states, indicating compatible intent and coordination patterns. Based on this compatibility, we exchange the corresponding blue agent segments between the two trajectories to create two new demonstrations (right). This operation preserves trajectory validity while enriching the diversity of collaborative behaviors.

behaviors of the blue agent in the two demonstrations are mutually acceptable—i.e., they follow a consistent collaboration pattern.

Given this compatibility, the remaining segments from the blue agent in the two trajectories can be exchanged. Replacing the blue agent's segment from demo A with that from demo B (and vice versa) produces two new valid demonstrations. Importantly, these recombined trajectories remain feasible within the environment while significantly increasing the diversity of collaborative behaviors represented in the dataset.

## P  TIME-CONTRASTIVE LEARNING AS A REWARD-FREE PROGRESS ESTIMATOR

To demonstrate that BASS does not require access to the environment reward, we further evaluate the behavior of the time-contrastive learning (TCL) Sermanet et al. (2017); Nair et al. (2022) model used as a reward-free progress estimator. TCL learns an embedding in which temporal ordering is preserved: earlier frames are embedded closer to the initial frame, while later frames progressively diverge. This structure allows TCL to provide a proxy measure of task progress using only state observations, enabling BASS to operate entirely within an imitation-learning setting.

Figure 14 shows the embedding distance between each frame of a trajectory and its first frame. The left panel presents five trajectories from the training set, and the right panel shows five trajectories from the test set. In both cases, the distances increase smoothly as time advances, indicating that TCL captures the underlying notion of task progression. Crucially, the model generalizes to unseen trajectories: the test curves follow patterns similar to the training curves, even though TCL was not trained on these sequences.

This consistency demonstrates that TCL provides a reliable and reward-free progress signal suitable for guiding action selection within BASS. Combined with the quantitative results in Table 16, these

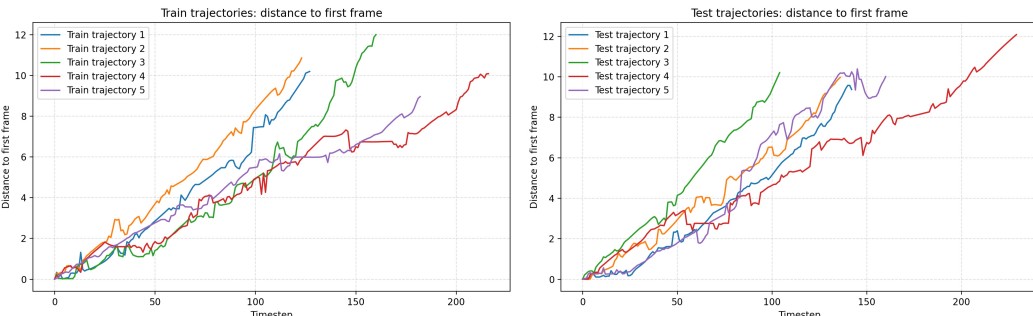

Figure 14: TCL progress estimation. Distances from each frame to the first frame for five training trajectories (left) and five test trajectories (right). The smooth and monotonic increase on both seen and unseen trajectories indicates that TCL provides a consistent and generalizable progress signal without requiring environment rewards.

Table 16: Performance of BASS using a time-contrastive learning (TCL) progress estimator on Moving Out Task 1 and Task 2.

| Method | Moving Out Task 1 | | | | Moving Out Task 2 | | | |
|---|---|---|---|---|---|---|---|---|
| | TCR(↑) | NFD(↑) | WT(↓) | AC(↑) | TCR(↑) | NFD(↑) | WT(↓) | AC(↑) |
| DP/BASS (Ours) | 0.3503 | 0.5724 | 0.3598 | 0.8337 | 0.4348 | 0.5535 | 0.3096 | 0.8474 |
| DP/BASS w/ reward estimation | 0.3448 | 0.5711 | 0.3475 | 0.8202 | 0.4328 | 0.5411 | 0.3187 | 0.8397 |
| DP | 0.3233 | 0.5367 | 0.3789 | 0.8163 | 0.3125 | 0.4526 | 0.3100 | 0.8442 |

findings confirm that the effectiveness of BASS does not rely on having access to environment rewards.

## Q COMPUTING RESOURCES

### Q.1 TRAINING

#### Q.1.1 BEHAVIOR CLONING

Models MLP and GRU are trained for 1000 epochs within approximately 0.5 to 1 hour on a single A6000 GPU. Training a diffusion policy, while also for 1000 epochs, generally requires a longer period of 1 to 3 hours. Overall, the computational time for behavior cloning methods is comparatively short.

#### Q.1.2 MAPPO

As MAPPO learns through direct interaction with the environment, it inherently requires a significantly greater number of training iterations. Currently, training MAPPO with 15 CPU threads typically spans 5 to 15 hours. Although MAPPO utilizes a lightweight MLP model with a small number of parameters, its training duration is extended due to two main factors:

- Firstly, the simulation environment, which is based on Pymunk, does not support GPU acceleration, thereby limiting the speed of physics calculations and environment stepping.
- Secondly, the computation of distance-based rewards becomes a bottleneck, particularly in environments featuring complex wall structures that necessitate more intensive calculations.

### Q.2 INFERENCE SPEED OF DP/BASS

Inference speed is critical for real-time human-AI collaboration, especially when interacting with human partners. In our setup, the environment runs at 10 Hz, i.e., each step occurs every 100 ms. While diffusion models are generally slower, our implementation generates the next 8 actions in 69 ms on an NVIDIA RTX A6000 GPU. This allows us to interact in real-time by predicting one

step in advance – at time step t, the agent executes the action predicted at t–1. This ensures smooth interaction without perceivable lags.

## R  DATA COLLECTION: TRAINING DATA

We conducted data collection for two tasks, each designed to evaluate different aspects of human-AI collaboration. For the two tasks, each participant controlled an agent using a joystick. The environment running at 10Hz for data collection.

For **Task 1**, which focuses on human behavior diversity, we recruited 36 participants, forming 18 groups of two. Before data collection, each group underwent a 10-minute practice session to familiarize itself with the environment. The remaining 50 minutes were dedicated to data collection. Each pair played each map three times, then switched agents and played three more times, resulting in six demonstrations per map. If a group completed all maps, they contributed a total of $12 \times 6 = 72$ human demonstrations. However, not all groups completed the full set, with some collecting only 3 to 5 demonstrations per map. Additionally, we removed low-quality demonstrations where performance was significantly poor. In total, we collected 1,000 valid human demonstrations for this task.

For **Task 2**, which evaluates adaptation to physical constraints, we worked with four expert players who were highly familiar with the environment. Each map had randomized object properties, ensuring variation in shape, size, and mass. Each map was played 60 times, resulting in $60 \times 12 = 720$ human demonstrations.

Our data collection and human study process was approved by an Institutional Review Board (IRB). Participants were compensated based on the amount of data they contributed, receiving between $15 to $20 per hour.

## S  HUMAN STUDY: PLAYING WITH MODELS

### S.1  HUMAN STUDY PROCEDURE

To collect data for our project, we designed an interactive experiment where human volunteers collaboratively played with trained AI agents. The data collection process is detailed as follows:

- **Model Selection:** Each volunteer was asked to select a model ID from four provided models (A, B, C, D).
- **Task Description and Limits:** After selecting a model, the volunteer played collaboratively with the AI agent across all twelve maps sequentially. The objective was to move all items on the map into the designated goal region. Each map had a time limit of 50 seconds. The volunteer could proceed to the next map either by successfully moving all items into the goal region or upon reaching the 50-second time limit.
- **Agent Roles:** For the first two models (A and B), the volunteer controlled the "red" agent while the AI controlled the "blue" agent. For the remaining two models (C and D), the roles were switched, with the volunteer controlling the "blue" agent and the AI taking the role of the "red" agent.
- **Questionnaire:** After completing all 12 maps for a given model, the volunteer filled out a questionnaire consisting of eight Likert-scale questions and one free-response question. Responses on the Likert scale ranged from "strongly agree" to "strongly disagree."

In total, we conducted this experiment with 12 volunteers. Each volunteer will be paid $20 for one hour of playing.

### S.2  QUESTIONNAIRE

We use the 7-Point Likert Scale for the questions below:

1. **Teamwork:** The other agent and I worked together towards a goal.
2. **Humanlike:** The other agent's actions were human-like.

3. **Reasonable:** The other agent always made reasonable actions throughout the game.

4. **Follow:** The other agent followed my lead when making decisions.

5. **Physics:** The other agent understands how to work with me when objects have varying physical characteristics.

6. **Helpfulness:** The other agent understands my intention and proactively helps me when I need assistance.

7. **Collision:** When our movement paths conflict, the other agent and I can effectively coordinate to avoid collisions.

8. **Alignment:** When moving large items together, our target directions remain well-aligned.

9. **Future:** I would like to collaborate with the other agent in future Moving Out tasks.

# T FULL RESULTS FOR TASK 1 AI-AI COLLABORATION

This section reports the complete experimental results for Task 1 under AI-AI collaboration.

| Evaluation Protocol | Method | TCR (↑) | TCR StdErr | NFD (↑) | NFD StdErr | WT (↓) | WT StdErr | AC (↑) | AC StdErr |
|---|---|---|---|---|---|---|---|---|---|
| Seen Behaviors | MLP | 0.2126 | 0.0072 | 0.2987 | 0.0048 | 0.4896 | 0.0021 | 0.8013 | 0.0093 |
| | GRU | 0.2369 | 0.0183 | 0.3011 | 0.0142 | 0.4975 | 0.0202 | 0.8151 | 0.0173 |
| | MAPPO | 0.1929 | 0.0038 | 0.3182 | 0.0045 | 0.5766 | 0.0068 | 0.8097 | 0.0071 |
| | DP | 0.3233 | 0.0279 | 0.5367 | 0.0151 | 0.3789 | 0.0167 | 0.8163 | 0.0162 |
| | DP/BASS | **0.3503** | 0.0293 | **0.5724** | 0.0232 | **0.3598** | 0.0182 | **0.8337** | 0.0146 |
| Unseen Behaviors | MLP | 0.1433 | 0.0061 | 0.2413 | 0.0033 | 0.5647 | 0.0031 | 0.7729 | 0.0090 |
| | GRU | 0.1638 | 0.0092 | 0.2453 | 0.0026 | 0.5758 | 0.0065 | 0.7830 | 0.0037 |
| | MAPPO | 0.1635 | 0.0067 | 0.2808 | 0.0037 | 0.6379 | 0.0013 | 0.7858 | 0.0050 |
| | DP | 0.2563 | 0.0152 | 0.4589 | 0.0177 | 0.4249 | 0.0136 | 0.7854 | 0.0041 |
| | DP/BASS | **0.3010** | 0.0223 | **0.5197** | 0.0361 | **0.3899** | 0.0245 | **0.8099** | 0.0179 |
| Play with Human | DP | 0.3855 | 0.0512 | 0.5547 | 0.0432 | 0.4886 | 0.0457 | 0.8054 | 0.0129 |
| | DP/BASS | **0.6512** | 0.0717 | **0.7053** | 0.0459 | **0.3364** | 0.0481 | **0.9124** | 0.0113 |

Table 17: Task 1 results under seen and unseen human behaviors, and with real human partners.

## U  RESULTS OF DIFFERENT METHODS WITH BASS

| Task 1 | TCR↑ | | NFD↑ | | WT↓ | | AC↑ | |
|---|---|---|---|---|---|---|---|---|
| Methods | Mean | Std Error | Mean | Std Error | Mean | Std Error | Mean | Std Error |
| MLP | 0.3568 | 0.0508 | 0.4118 | 0.0338 | 0.4380 | 0.0419 | 0.7890 | 0.0250 |
| + BASS w/o Simulation | 0.2952 | 0.0436 | 0.4207 | 0.0359 | 0.3639 | 0.0358 | 0.8060 | 0.0126 |
| GRU | 0.3070 | 0.0479 | 0.3674 | 0.0365 | 0.3143 | 0.0532 | 0.7618 | 0.0201 |
| + BASS w/o Simulation | 0.4117 | 0.0465 | 0.4396 | 0.0350 | 0.3891 | 0.0418 | 0.8225 | 0.0173 |
| + BASS w/o Augmentation | 0.3531 | 0.0411 | 0.4047 | 0.0373 | 0.3835 | 0.0419 | 0.8195 | 0.0210 |
| + Full BASS | 0.4120 | 0.0513 | 0.4454 | 0.0392 | 0.4218 | 0.0426 | 0.8345 | 0.0173 |
| Diffusion Policy (DP) | 0.3829 | 0.0681 | 0.4818 | 0.0514 | 0.3075 | 0.0374 | 0.8395 | 0.0216 |
| + BAAS w/o Simulation | 0.4028 | 0.0666 | 0.5114 | 0.0493 | 0.3392 | 0.0428 | 0.8242 | 0.0254 |
| + BASS w/o Augmentation | 0.4741 | 0.0667 | 0.5561 | 0.0506 | 0.3176 | 0.0435 | 0.8495 | 0.0174 |
| + Full BASS | 0.5027 | 0.0619 | 0.5707 | 0.0468 | 0.3448 | 0.0402 | 0.8615 | 0.0167 |

Table 18: The table presents all experimental results for Task 1 in seen behaviors.

| Task 2 | TCR↑ | | NFD↑ | | WT↓ | | AC↑ | |
|---|---|---|---|---|---|---|---|---|
| Methods | Mean | Std Error | Mean | Std Error | Mean | Std Error | Mean | Std Error |
| MLP | 0.2557 | 0.0413 | 0.3602 | 0.0315 | 0.4867 | 0.0418 | 0.8175 | 0.0261 |
| + BASS w/o Simulation | 0.2014 | 0.0336 | 0.3656 | 0.0244 | 0.3657 | 0.0332 | 0.7890 | 0.0250 |
| GRU | 0.2582 | 0.0509 | 0.3935 | 0.0428 | 0.4680 | 0.0594 | 0.8487 | 0.0183 |
| + BASS w/o Simulation | 0.3333 | 0.0539 | 0.4141 | 0.0439 | 0.5611 | 0.0587 | 0.8513 | 0.0286 |
| + BASS w/o Augmentation | 0.3670 | 0.0522 | 0.4246 | 0.0420 | 0.4365 | 0.0593 | 0.8572 | 0.0222 |
| + Full BASS | 0.3414 | 0.0522 | 0.4410 | 0.0442 | 0.4379 | 0.0596 | 0.8754 | 0.0165 |
| Diffuson Policy (DP) | 0.3125 | 0.0564 | 0.4526 | 0.0427 | 0.3100 | 0.0385 | 0.8442 | 0.0184 |
| + BAAS w/o Simulation | 0.3569 | 0.0547 | 0.4908 | 0.0385 | 0.3256 | 0.0431 | 0.8373 | 0.0147 |
| + BASS w/o Augmentation | 0.4200 | 0.0544 | 0.5187 | 0.0417 | 0.3232 | 0.0417 | 0.8305 | 0.0169 |
| + Full BASS | 0.4348 | 0.0599 | 0.5535 | 0.0423 | 0.3096 | 0.0451 | 0.8474 | 0.0128 |

Table 19: The table presents all experimental results for Task 2

## V  MAP ANALYSIS

Our 12 maps are carefully designed to target specific collaboration modes, coordination, awareness, and action consistency, while ensuring that existing AI agents (e.g., MAPPO, Diffusion Policy) can perform some tasks but still exhibit clear limitations. This balance is essential: overly difficult maps with long paths or dense obstacles may yield near-zero performance for all agents, making it impossible to evaluate various aspects of human-AI collaboration. Our map definition is already based on a structural format, e.g., JSON, allowing easy modification, reuse of modules, and procedural generation for scalability.

### V.1  COORDINATION

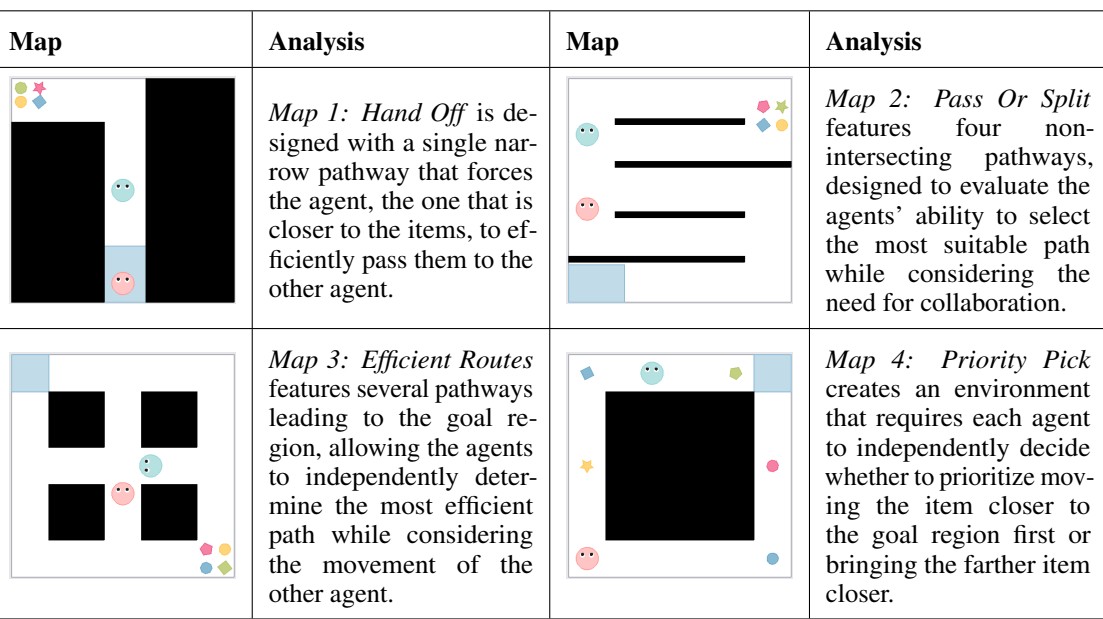

| Map | Analysis | Map | Analysis |
|---|---|---|---|
| | *Map 1: Hand Off* is designed with a single narrow pathway that forces the agent, the one that is closer to the items, to efficiently pass them to the other agent. | | *Map 2: Pass Or Split* features four non-intersecting pathways, designed to evaluate the agents' ability to select the most suitable path while considering the need for collaboration. |
| | *Map 3: Efficient Routes* features several pathways leading to the goal region, allowing the agents to independently determine the most efficient path while considering the movement of the other agent. | | *Map 4: Priority Pick* creates an environment that requires each agent to independently decide whether to prioritize moving the item closer to the goal region first or bringing the farther item closer. |

Table 20: Maps categorized under **Coordination**.

V.2 AWARENESS

| Map | Analysis | Map | Analysis |
|---|---|---|---|
| | *Map 5: Corner Decision* requires the agents to decide whether to follow the other agent to the upper right or the lower left corner and to determine which size of item to prioritize moving first. | | *Map 6: Distance Priority* contains two medium-sized items, requiring the agents to decide whether to prioritize the item that is farther away or the one that is closer. |
| | *Map 7: Top Bottom Priority* contains two items, either large or medium-sized, requiring the agents to decide whether to prioritize the item at the top or the one at the bottom. | | *Map 8: Adaptive Assist* contains a mix of large or medium-sized items and small items, requiring the agents to decide whether to prioritize collaborating on the larger item or individually handling the smaller item. |

Table 21: Maps categorized under **Awareness**.

## V.3 ACTION CONSISTENCY

| Map | Analysis | Map | Analysis |
|---|---|---|---|
|  | *Map 9: Left Right* contains large-sized items, requiring the agents to continuously collaborate and make strategic decisions about whether to move items to the left or right goal region. |  | *Map 10: Single Rotation* contains one large-sized item, which is designed to evaluate how well the two agents can collaborate to perform a single rotation. |
|  | *Map 11: Four Corners* contains large-sized items positioned at the four corners, requiring the agents to continuously collaborate by moving the items in either a clockwise or counterclockwise order. |  | *Map 12: Sequential Rotations* contains one large-sized item, which is designed to evaluate how well the two agents can collaborate to maintain a sequence of rotations. |

Table 22: Maps categorized under **Action Consistency**.

