# OpenReview forum: "Moving Out: Physically-grounded Human-AI Collaboration"
_ICLR.cc/2026/Conference — Submitted to ICLR 2026_

### Official Review · Reviewer_68az · 2025-10-29

**Soundness:** 2
**Presentation:** 2
**Contribution:** 2
**Rating:** 4
**Confidence:** 4

**Summary:**

This paper introduces Moving Out, a physically grounded benchmark for human-AI collaboration that emphasizes continuous action control and physical constraints. The benchmark includes two tasks supported by over 1,700 demonstrations. To address these challenges, the authors propose BASS, which augments collaborative trajectories, and simulates next states via a dynamics model under physical constraints. Experiments demonstrate that BASS outperforms other baselines in both AI-AI and human-AI collaboration, achieving higher task completion rates and better coordination.

**Strengths:**

1. This paper contributes a well-designed benchmark for symbolic human-AI collaboration and physically grounded interaction.

2. The BASS method integrates data augmentation and next-state prediction to enhance robustness to diverse human behavior and physical variations.

3. Extensive experiments, such as ablation studies, human user studies, and failure case analyses validate performance improvements.

**Weaknesses:**

1. Some key terms are not clearly defined, which reduces the overall clarity of the paper.

- L11: The phrase “adapt to physical actions and constraints” lacks precise definition. The authors should clarify what constitutes “physical actions” and “constraints.”

- L42: The scope of “diverse physical constraints, variations, and behavior” should be specified more precisely.

- L339, L341: The meanings of “aligned the boundaries” and “match the boundaries” are unclear.

- Title, Fig. 1, etc.: The authors emphasize a “physically grounded setting,” but the environments appear to be interactive simulations with limited rule sets (e.g., object mass, movement logic). The distinction between “physically grounded” and “grid world” is not well defined. If the intention is to highlight continuous action spaces, this should be stated explicitly.

2. The benchmark includes only two tasks (behavioral and physical generalization) with two agents. The limited diversity of agents and task types makes it unclear whether the benchmark captures broader aspects of collaboration, such as communication, planning, or anticipation.

3. In the Behavior Augmentation module, the recombination of sub-trajectories seems to involve exchanging trajectory segments between agents. The authors should clarify how this process improves action consistency.

4. Although the ablation study shows performance gains, it remains unclear how each submodule of BASS (augmentation, simulation, and selection) contributes to performance under different physical conditions or tasks.

5. The paper provides limited comparison with recent embodied collaboration frameworks. While HumanTHOR and Habitat 3.0 are mentioned, direct experimental comparisons are missing, which weakens the practical significance of the proposed approach.

**Questions:**

1. Some key terms are expected for clear definitions.

2. How do the proposed environments differ fundamentally from typical “grid worlds”? What specific properties make them “physically grounded”?

3. Given that the benchmark includes only two agents and two task types, do these settings generalize to broader aspects of human–AI collaboration, such as communication or planning?

4. How does sub-trajectory recombination improve action consistency between agents?

5. How each BASS submodule (augmentation, simulation, selection) contributes to performance under different conditions?

6. More collaboration frameworks such as HumanTHOR or Habitat 3.0 are expected to be compared.

---

> ### Author Response · Authors · 2025-11-24
>
> We thank the reviewer for the detailed feedback. We appreciate the
> recognition of the benchmark design, the integration of BASS components,
> and the breadth of our empirical evaluation. The reviewer's comments on
> clarity, terminology, task scope, and comparisons with related
> frameworks are very valuable, and we address each point in the following
> response.
>
> ## W1, Q1 clarification on physical related terms
>
> ### Symbolic human-AI collaboration
>
> In Strengths 1, the description suggests that our setting is a symbolic
> human-AI collaboration. We would like to clarify that the symbolic
> setting applies to the baselines we compare against, such as
> Overcooked-AI and similar discrete environments. Our benchmark, while
> the objects to transport can be represented symbolically, is primarily
> built on a continuous state-action space and operates in a physically
> grounded setting powered by a physics engine.
>
> ### "Physical actions" and "physical constraints"
>
> Our environment is physically grounded because it is simulated using a
> PyMunk physics engine rather than a grid world. By "physical actions,"
> we refer to continuous control inputs that produce continuous motions
> under physical laws, rather than symbolic tasks (e.g., grasp an onion)
> or discrete moves (e.g., moving one grid cell or moving 5 cm to the
> right). Agents receive continuous force actions, experience contact
> responses, and must handle friction, collision, mass, shape, and
> rotation effects.
>
> "Physical constraints" refer to environmental and object-level
> restrictions such as object mass affecting acceleration during carrying,
> object geometry requiring different grasping or rotation strategies to
> pass through doors, and collision-based limitations that must be
> resolved through continuous action adjustment. We will clarify these
> definitions in the revision.
>
> ### The scope of diverse physical constraints, variations, and behavior
>
> In the Overcooked AI (the grid world shown in Figure 1), each agent has
> only three possible positions near the counter, and there are only three
> designated locations where items can be handed over. Because the state
> space is limited and highly structured, human behavior diversity can be
> covered easily through randomization, making it difficult to
> meaningfully evaluate "unseen human behavior." In contrast, our
> environment is fully continuous and physics-driven. The state space is
> infinite; it is almost impossible for two human demonstrations to
> produce identical states. This allows us to study a key challenge in
> human-AI collaboration: adapting to unseen and continuous behavior
> variations that do not appear in the training set.
>
> Similarly, physical variations in grid worlds are symbolic (e.g., onion
> vs tomato), but the actions required to manipulate them are identical.
> In our environment, physical variation affects dynamics directly: mass
> changes motion, friction affects pushing, and shape influences how
> agents must coordinate when rotating or maneuvering an object. We have
> updated the introduction in the revised version.
>
> ### "Aligned boundaries\" and "matched boundaries\"
>
> We provided a visualization example in Appendix O.1 to easier to understand
> the recombination process.
>
> We identify a segment of agent $i$ in demonstration A between timesteps
> $t_1$ and $t_2$, and a segment in demonstration B between $t_3$ and
> $t_4$, where the agent begins and ends in the same observable physical
> situation. Here, "the same" does not require exact equality because a
> continuous environment rarely produces identical states. Instead, we
> treat two states as the same if the agent's pose differs by less than
> $1.25\%$, which is a small tolerance relative to the scale of the
> environment and visually indistinguishable in practice. Formally, this
> condition is expressed as $s^i_{t_1} \approx \hat{s}^i_{t_3}
> %\qquad
> \text{and}
> %\qquad
> s^i_{t_2} \approx \hat{s}^i_{t_4},$ with $t_2 > t_1$ and $t_4 > t_3$.
> This requirement is what we mean by "aligned boundaries\": The start
> states match and the end states match between two demonstrations for
> agent $i$. It ensures that agent $i$ enters and exits the segment from
> the same physical situation, even though the segment occurs at different
> times in the two trajectories.
>
> "Matching the boundaries\" refers to the same condition, but viewed as a
> constraint in the recombination of agent $j$'s trajectories: Only when
> agent $i$ is in the same states (pose and holed items) at the beginning
> and end of the two segments do we treat them as compatible. Under
> matched boundaries, we can swap partner $j$'s sub-trajectories,
> $\tau^j_{t_1:t_2}  \longleftrightarrow\ \hat{\tau}^j_{t_3:t_4},$ while
> keeping agent $i$'s movement unchanged. Because agent $i$ starts and
> ends the segment in the same states in both demonstrations, the
> recombined trajectory remains coherent and physically valid, with only
> the partner's behavior differing. We also updated Sec. 5.1 to clarify
> the algorithm for recombination.

---

> > ### Author Response · Authors · 2025-11-24
> >
> > ## Q2 Physically-grounded
> >
> > We clarify that continuous state and action spaces are only the
> > foundation of a physically grounded environment, but continuous
> > state-action alone does not define physical grounding. By "physically
> > grounded," we refer to tasks that are affected by physical attributes
> > and need to satisfy physical constraints. These two properties match the
> > challenges we discuss in Sec. 3, Problem Definition. This definition is
> > similar to real-world robotic scenarios, for example, in a warehouse
> > scenario, objects differ in size, mass, shape, and inertia. Our
> > environment uses a physics engine to model these variations, and agents
> > must learn to handle different physical properties through continuous
> > control. This is distinct from grid-world environments, where
> > interactions with objects are symbolic and homogeneous. We have updated
> > the introduction in the revised paper.
> >
> > ## W2, Q3 Task diversity and broader aspects
> >
> > Our benchmark includes two tasks that are intentionally designed. As
> > Reviewer 3JQP noted, environments can always be made more complex, but
> > higher complexity does not necessarily help us understand the core
> > challenges in human-AI collaboration. The two problems we focus on,
> > generalization to unseen human behaviors and generalization to unseen
> > physical attributes, are central to all subsequent aspects of
> > collaboration and widely recognized challenges in the field. A more
> > complex environment may further stress-test AI models, but our current
> > task complexity/diversity carefully balances the task difficulty and
> > collaboration challenges. We instantiate the environment to allow
> > existing agents to make partial progress, but hard to complete without
> > effective collaboration, so we can focus on studying human-AI
> > interaction rather than improving an agent's task capacity.
> >
> > While we intentionally do not include explicit communication and
> > planning in our benchmark to focus on the effects of two core aspects
> > (behavior diversity and physical variation), it is possible to
> > generalize the settings by adding a planning stage and a communication
> > channel. These additions will introduce additional factors that
> > influence collaboration performance, e.g., computation budget for
> > planning and types/frequency of messages in communication, and are
> > important future work. It is also worth noting that our current
> > benchmark employs implicit communication that naturally arises from the
> > physical states and motion of the agents, such as visible slow progress
> > when a large object requires assistance. The waiting time metrics can be
> > a proxy for the effectiveness of this implicit communication. This
> > allows us to study human-AI collaboration without confounding effects
> > from language understanding. We have updated this part in the revised
> > paper, section 4.1.2.
> >
> > ## W3, Q4 How recombination improves action consistency
> >
> > In our method, behavior augmentation is designed to increase the
> > diversity of partner behaviors rather than directly optimizing an
> > action-consistency metric. If the reviewer is referring to consistency
> > in the sense of preserving the agent's intended goal within the
> > trajectory, this is ensured through the way we perform multi-agent
> > sub-trajectory recombination. A segment is only exchanged when the agent
> > is in a comparable state and follows the same intermediate objective in
> > both demonstrations, so the recombined trajectory maintains coherent
> > intent while still enriching the diversity of collaborative behaviors.
> >
> > As described in Appendix O, each trajectory is decomposed into the
> > states of the two agents and the environment. Recombination is applied
> > only when an agent exhibits a segment that begins from nearly the same
> > state in two demonstrations. When the starting situation and behavioral
> > intent of this segment match, the partner's corresponding segment
> > becomes interchangeable across demonstrations.
> >
> > This compatibility requirement ensures that the agent maintains the same
> > behavioral intention over the swapped interval, while only the partner's
> > behavior varies. The resulting recombined trajectory remains coherent
> > and physically plausible, increasing behavioral diversity without
> > producing contradictory or invalid joint behaviors.

---

> > > ### Author Response · Authors · 2025-11-24
> > >
> > > ## W4, Q5 How does each BASS submodule contribute under different conditions?
> > >
> > > We already included ablation studies and the module-level analyses in
> > > Appendix L.1. Additionally, we conducted new experiments comparing the
> > > full BASS system with single-agent variants and include the full results
> > > in Appendix L.2. The analysis shows the contribution of each BASS
> > > submodule under different physical conditions and across both tasks.
> > > Below, we summarize the main findings.
> > >
> > > ### Behavior Augmentation.
> > >
> > > To isolate the effect of augmentation, we compare our multi-agent
> > > recombination with a single-agent behavior augmentation baseline that
> > > ignores alignment between the two agents. In the single-agent
> > > recombination setting, a segment is considered compatible as long as
> > > either its start or end state matches another trajectory, that is,
> > > $s^i_{t_1} \approx \hat{s}^i_{t_3}
> > > \text{or}
> > > s^i_{t_2} \approx \hat{s}^i_{t_4},$ with $t_2 > t_1$ and $t_4 > t_3$,
> > > because the method does not enforce consistency in the agent's
> > > behavioral goal. As a result, these segments may merely overlap at a
> > > point rather than represent the same intended behavior, and recombining
> > > them cannot ensure goal alignment in a multi-agent collaborative
> > > setting.
> > > As shown in the table below, the single-agent variant produces
> > > higher entropy $(0.897)$ but reduces collaboration performance, lowering
> > > TCR and NFD from $(0.403,\,0.511)$ to $(0.368,\,0.451)$. This occurs
> > > because the swapped partner segment often corresponds to a different
> > > state of the two agents and the shared object, so the combined motions
> > > no longer produce a physically consistent joint action required for
> > > collaboration. In contrast, our multi-agent recombination only swaps
> > > segments when the two demonstrations place the agent $i$ in the same
> > > states at the start and end of the segment, which ensures the coherent
> > > joint behavior. This result shows that augmentation contributes only
> > > when the two agents' motions remain compatible after recombination.
> > >
> > > | Method                         | DTW Mean | DTW Var | Entropy (KDE) | Coverage (RBF) | TCR ↑  | NFD ↑  | WT ↓   | AC ↑   |
> > > |--------------------------------|----------|---------|----------------|----------------|--------|--------|--------|--------|
> > > | **Multi-agent Recombination (Ours)** | 7.526    | 6.612   | 0.892          | 0.910          | **0.403** | **0.511** | **0.308** | **0.840** |
> > > | **Single-agent Recombination**       | **7.741** | **6.722** | **0.897**        | **0.919**        | 0.368  | 0.451  | 0.338  | 0.839  |
> > > | **No Recombination**                 | 7.013    | 6.065   | 0.888          | 0.899          | 0.383  | 0.482  | 0.345  | 0.824  |
> > >
> > >
> > > ### Simulation and Action Selection.
> > >
> > > Simulation and selection are coupled, so we analyze them together. The
> > > single-agent simulation variant predicts only one agent's future state
> > > and ignores how the partner will move.
> > > As shown in the table below, this significantly reduces TCR
> > > and NFD from $(0.420,\,0.554)$ in the multi-agent version to
> > > $(0.319,\,0.458)$, which is close to disabling simulation entirely
> > > $(0.313,\,0.452)$. Without modeling the partner's future motion or the
> > > coupled dynamics of physical manipulation, the system cannot reliably
> > > judge whether a candidate action will contribute to progress or cause
> > > blocking.
> > >
> > > | Method                             | TCR ↑    | NFD ↑    | WT ↓     | AC ↑     |
> > > |------------------------------------|----------|----------|----------|----------|
> > > | **Multi-agent Action Simulation (Ours)** | **0.420** | **0.554** | **0.310** | **0.848** |
> > > | **Single-agent Action Simulation**       | 0.319    | 0.458    | 0.327    | 0.833    |
> > > | **No Action Simulation**                 | 0.313    | 0.452    | 0.335    | 0.821    |
> > >
> > >
> > > Additional analysis in Appendix L.1 supports the same conclusion. The
> > > dynamics module achieves an L2 error of $0.0010$ and $0.0028$ in Tasks 1
> > > and 2, far better than a GRU predictor or random states. Partner action
> > > prediction reaches $71.45\%$ accuracy under a 10% tolerance. When
> > > replacing the partner model with random actions, both action-selection
> > > accuracy and TCR drop sharply. These results show that simulation and
> > > selection contribute to predicting joint future motion and selecting
> > > actions that maintain collaborative progress under diverse physical and
> > > behavioral conditions.
> > >
> > > ### Summary.
> > >
> > > Across tasks and physical variations, augmentation improves robustness
> > > to diverse partner behaviors only when cross-agent compatibility is
> > > enforced, and simulation plus selection enable BASS to choose actions
> > > that avoid blocking and promote progress. The single-agent comparisons
> > > demonstrate that each BASS submodule is effective. More detailed
> > > analysis is provided in Appendix L.

---

> > > > ### Author Response · Authors · 2025-11-24
> > > >
> > > > ## W5, Q6 Comparison with HumanThor and Habitat 3.0
> > > >
> > > > ### Collaboration Modes
> > > >
> > > > Compared with HumanTHOR and Habitat 3.0, Moving Out supports a
> > > > substantially broader set of collaboration modes. Both prior platforms
> > > > focus on division-of-labor tasks such as navigation and rearrangement.
> > > > They do not support the physically coupled collaboration modes shown in
> > > > our benchmark, such as jointly carrying or rotating objects. These tasks
> > > > require real-time synchronous action and continuous low-level control.
> > > >
> > > > ### Datasets and Evaluation
> > > >
> > > > Neither HumanTHOR nor Habitat provides datasets or challenge-driven task
> > > > suites for studying human-AI collaboration under diverse human behaviors
> > > > and diverse physical constraints. They also do not provide dedicated
> > > > tasks or evaluation protocols for challenges in studying unseen
> > > > behaviors or unseen physical constraints.
> > > >
> > > > ### Scope and Intended Use
> > > >
> > > > The scopes of HumanTHOR and Habitat are fundamentally different from
> > > > Moving Out. HumanTHOR does not train or evaluate AI agents and relies
> > > > solely on rule-based heuristics such as frontier exploration and oracle
> > > > detectors. Habitat supports low-level RL control mainly for navigation,
> > > > but its rearrangement tasks rely on predefined or pre-trained low-level
> > > > skills policies. As a result, Habitat cannot provide low-level policies
> > > > for physically multi-agent collaboration, whereas our benchmark allows
> > > > learning continuous low-level control for joint manipulation tasks.
> > > > These settings primarily assess spatial understanding and planning
> > > > rather than physical human-AI cooperation with low-level motion. In
> > > > contrast, Moving Out provides continuous low-level control throughout
> > > > the entire collaboration process, making it purposefully designed for
> > > > studying physically grounded teamwork.
> > > >
> > > > ### Summary.
> > > >
> > > > Overall, Moving Out fills a critical gap left by existing embodied
> > > > collaboration frameworks. It offers physically grounded tasks,
> > > > continuous control, real human behavior data, and challenge-based
> > > > evaluations that explicitly reveal the limitations of current AI systems
> > > > in human-AI collaboration, whereas HumanTHOR and Habitat focus on
> > > > navigation or scripted manipulation and lack of datasets.
> > > >
> > > > We hope that our responses address the reviewer's concerns, and we would
> > > > be very happy to provide any further clarification or discussion if
> > > > needed.

---

### Official Review · Reviewer_3JQP · 2025-10-31

**Soundness:** 3
**Presentation:** 3
**Contribution:** 3
**Rating:** 6
**Confidence:** 3

**Summary:**

This paper presents Moving Out, a human-AI collaboration benchmark, and BASS, a method for enhancing agent capability to handle human collaboration. The paper begins by motivating this problem and approach through the need for agents that can adapt to unseen human behaviors in collaborative settings. It outlines the high-level collaborations of a two-task benchmark, demonstration data for each task, and a novel data/training recipe for enhancing human-AI collaboration ability, consisting of components trained on this data.

The paper goes on to formalize the problem as a decentralized MDP with two agents and discuss challenges of collaboration - unknown human policy distribution and large joint action and observation spaces. It then describes the environment and tasks. In the first task, focused on predicting and handling unseen human behaviors, models are trained on human-human collaboration demos, then tested with a new collaborator to see if they can generalize past seen behaviors. Evaluation is done by training multiple agents on splits of the data, then having the trained models collaborate with each other. The second task tests handling of physical attributes, particularly variation; evaluation is done by training agents on the full dataset and having them self-play. Demonstrations are collected from real humans as part of this work.

The paper then describes the augmentation method BASS. Partner poses are perturbed and subtrajectories are recombined to maintain coherence. A three-stage pipeline is built: an autoencoder to encode states, a dynamics model to predict the next latent state, and a second autoencoder to reconstruct the predicted state. Actions are chosen based on highest reward, where reward is a function of object distances to the goal region. Agents can then be trained on this data and select actions according to this framework.

Finally, the paper reports on experiments. It poses four research questions: does BASS adapt to unseen behaviors, does BASS generalize to unseen physical constraints, does BASS improve effectiveness of collaboration, and failure patterns of BASS. Results show that Diffusion Policy (DP) with BASS is able to maintain the most performance across methods, though some others come close; DP with BASS outperforms baselines on certain key metrics measuring physical generalization; DP with BASS shows strong improvement over DP on human-collaborative tasks; and failure modes include various aspects of adapting to diverse behaviors, highlighting the inherent difficulty of high-level, conceptually diverse stimuli.

**Strengths:**

### Quality
- Paper is very well-motivated
- RQ structure is thoughtful. I appreciate not just that there are RQs, but that they target so many aspects of one concept, including failure modes.
### Clarity
- Well written
- Well-designed figures
- Research question structure is very useful - helps understand what the paper is arguing. This is even more important given that the tasks, envs, data, etc. are expansive and somewhat arbitrary. Without the RQs and with just numbers, it would be hard to ascribe meaning to the results.
### Originality and significance
BASS seems novel, and the methods for constructing various aspects of the data are innovative and clever.


Overall, I think the benchmark is an interesting and useful contribution, even if there are related datasets/collection strategies and the design is similar to others. BASS is novel to my knowledge, but not particularly surprising given its reliance on the reward. Regardless, however, this paper is useful, and I think that warrants acceptance.

**Weaknesses:**

### Quality
- Design is somewhat arbitrary. The tasks are very general, but their instantiations - the objects, environment, domain expansion techniques, etc. - are specific and feel somewhat arbitrary. Ultimately that might be fine, as all benchmarks have to be feasible to build and use, but it would help to have some justification for the particular instantiations.
- Evaluation lacks control. The numbers are convincing - BASS shows improvement on the tasks, and the tasks are large and expansive enough that I buy that that is meaningful. However, the evaluation is very high-level. Training AIs and then having them collaborate with humans, other AIs, or other AIs trained on this data - the behavior of such systems can have many explanations that are not necessarily differentiated by these experiments.
- The results are strong, but they do rely on the reward signal, which adds an explicit addition over standard imitation learning. It's not surprising that that information helps.
### Clarity
- No major concerns on clarity - the paper is put together well.
### Originality and significance
Human-AI collaboration is an established field. While BASS and the experiments done on it seem novel, the dataset is a very useful artifact but I'm not sure it's entirely novel. That said, since it is unique and useful, I don't think this is a major issue.

**Questions:**

I agree that the subtrajectory recombination is coherent. However, I suspect it deviates from naturalism. This may not matter - you don't need naturalism to have a good benchmark of complex behaviors, and it often only serves to add further complexity - but how do we think about the resulting distribution? What are some examples of pre- and post-recombination trajectories that build intuition for how/why BASS works?

---

> ### Author Response · Authors · 2025-11-24
>
> We thank the reviewer for the detailed and thoughtful assessment of our
> work. We greatly appreciate the recognition of the motivation behind
> Moving Out, the usefulness of our benchmark, the clarity of the
> RQ-driven structure, and the novelty and practicality of the BASS
> framework. We address the remaining questions below.
>
> ## W1 Concern about the design being arbitrary; design choices
>
> Our domain and environment designs are intentional. Here, we explain the
> rationale behind the instantiation in Moving Out.
>
> **Task Domain**: Our selected domain, collaborative
> carrying/manipulation, is motivated by real human-robot collaboration
> where robots operate in continuous state and action spaces and must
> consider physical properties such as mass, shape, and friction. A
> similar setup is common in real applications, such as warehouse
> operations and industrial/service scenarios. In such scenarios, the
> difficulty of collaboration is affected by the physical attributes of
> objects and the environment layouts. We selected three types of items: a
> small object that one agent can move alone, a medium object that one
> agent can move but performs better with help, and a large object that
> strongly requires cooperation. These choices reflect practical use cases
> and allow us to study different forms of collaboration in a physically
> grounded setting. We believe these instantiations are meaningful and
> motivated by real-world considerations.
>
> **Environment**: The instantiation of the maps, including types and
> placements of objects, environment layouts, and spawn points of agents,
> is chosen via repeated experiments with multiple versions of the
> environment such as varying number of obstacles and size of open space.
> The final instantiations are selected to balance the collaboration modes
> and task complexity. Specifically, we ensured that the task allows baseline RL and IL agents to make partial progress, yet they fail to collaborate effectively with other agents. Ensuring that baseline agents exhibit reasonable behavior allows us to focus on studying their shortcomings in human-AI coordination under unseen human behaviors and physical constraints, rather than simply evaluating the basic capability of RL/IL algorithms.
>
> ## W2 Controls in evaluations
>
> We agree that controlled evaluation is important. In addition to the
> system-level experiments in the main paper, we wanted to point out that
> we already included module-level ablations in Appx. C, where we had
> ablations in behavior augmentation and action simulation and selection,
> showing that each component contributes to performance.
>
> To further address the reviewer's concern, we have added more controlled
> comparisons. First, for behavior augmentation, we compare our
> multi-agent recombination with a single-agent recombination variant that
> ignores partner alignment. The table below shows single-agent recombination
> produces higher diversity scores but leads to worse collaboration
> performance, because the generated trajectories no longer maintain the
> coordinated goals of both agents. This demonstrates that our
> augmentation effectively handles multi-agent consistency, which is
> required for effective collaboration.
> | Method                         | DTW Mean | DTW Var | Entropy (KDE) | Coverage (RBF) | TCR ↑  | NFD ↑  | WT ↓   | AC ↑   |
> |--------------------------------|----------|---------|----------------|----------------|--------|--------|--------|--------|
> | **Multi-agent Recombination (Ours)** | 7.526    | 6.612   | 0.892          | 0.910          | **0.403** | **0.511** | **0.308** | **0.840** |
> | **Single-agent Recombination**       | **7.741** | **6.722** | **0.897**        | **0.919**        | 0.368  | 0.451  | 0.338  | 0.839  |
> | **No Recombination**                 | 7.013    | 6.065   | 0.888          | 0.899          | 0.383  | 0.482  | 0.345  | 0.824  |
>
> Second, we evaluate the action simulation and selection module in
> isolation. We compare our multi-agent dynamics model with a single-agent
> variant that predicts only one agent's future state without modeling the
> partner's motion. The table below shows the single-agent version performs
> similarly compared to disabling simulation entirely, whereas the
> multi-agent model provides substantial gains. This indicates that
> accurate simulation requires capturing the coupled evolution of both
> agents and the design in BASS captures this aspect.
>
> | Method                             | TCR ↑    | NFD ↑    | WT ↓     | AC ↑     |
> |------------------------------------|----------|----------|----------|----------|
> | **Multi-agent Action Simulation (Ours)** | **0.420** | **0.554** | **0.310** | **0.848** |
> | **Single-agent Action Simulation**       | 0.319    | 0.458    | 0.327    | 0.833    |
> | **No Action Simulation**                 | 0.313    | 0.452    | 0.335    | 0.821    |
>
> The controlled comparisons above provide further evidence that the
> improvements from BASS come from increased behavior diversity and more
> accurate action selection.

---

> ### Author Response · Authors · 2025-11-24
>
> ## W3 The results rely on reward signals
>
> The current version of BASS indeed uses the environment reward to score
> candidate actions. However, BASS can also work without access to the
> reward signal. To directly address the reviewer's concern, we include an
> additional experiment showing that BASS remains effective in a fully
> imitation-learning setting. Specifically, we replace the environment
> reward with a progress estimator trained using time-contrastive learning
> (TCL). \[1\] TCL learns an embedding space where temporally close
> demonstration frames map to nearby representations while distant frames
> map farther apart, enabling the model to estimate task progress, and
> thus a proxy reward, purely from pairs of observed states. As shown in the table below, BASS maintains strong performance under
> this reward-free setup, demonstrating that the framework does not
> necessarily rely on environment rewards. The corresponding
> visualizations have been added to Appendix P.
>
> | Method | **Moving Out Task 1** | | | | **Moving Out Task 2** | | | |
> | :--- | :--- | :--- | :--- | :--- | :--- | :--- | :--- | :--- |
> | | **TCR(↑)** | **NFD(↑)** | **WT(↓)** | **AC(↑)** | **TCR(↑)** | **NFD(↑)** | **WT(↓)** | **AC(↑)** |
> | **DP/BASS (Ours)** | 0.3503 | 0.5724 | 0.3598 | 0.8337 | 0.4348 | 0.5535 | 0.3096 | 0.8474 |
> | **DP/BASS w/ reward est.** | 0.3448 | 0.5711 | 0.3475 | 0.8202 | 0.4328 | 0.5411 | 0.3187 | 0.8397 |
> | **DP** | 0.3233 | 0.5367 | 0.3789 | 0.8163 | 0.3125 | 0.4526 | 0.3100 | 0.8442 |
>
> \[1\] Pierre Sermanet, Kelvin Xu, and Sergey Levine. Unsupervised
> perceptual rewards for imitation learning. RSS 2017.
>
> We hope that our responses address the reviewer's concerns, and we would
> be very happy to provide any further clarification or discussion if
> needed.

---

### Official Review · Reviewer_EuNa · 2025-10-31

**Soundness:** 3
**Presentation:** 3
**Contribution:** 2
**Rating:** 6
**Confidence:** 2

**Summary:**

This paper presents Moving Out, a benchmark for physically grounded human-AI collaboration that addresses the challenges of continuous state-action spaces and physical constraints such as mass, shape, and friction. It introduces two core tasks—adapting to diverse human behaviors and generalizing to unseen physical attributes—and proposes BASS (Behavior Augmentation, Simulation, and Selection), a framework that integrates sub-trajectory swapping for data augmentation, a learned dynamics model for next-state prediction, and an action-selection mechanism to optimize actions under physical constraints. Extensive experiments and human-subject studies show that BASS significantly enhances task completion, action consistency, and robustness compared to strong baselines including GRU, MAPPO, and Diffusion Policy.

**Strengths:**

The Moving Out environment bridges the gap between symbolic multi-agent environments (e.g., Overcooked-AI) and physically grounded continuous control, with explicit physics and multi-modal collaboration types (coordination, awareness, action consistency).

The BASS framework introduces behavior recombination for partner diversity and a next-state simulation module for physical reasoning, which is well-motivated and conceptually solid.

Quantitative and qualitative analyses (Task Completion Rate, Normalized Final Distance, Waiting Time, Action Consistency) show clear improvements over competitive baselines.

**Weaknesses:**

The augmentation and simulation modules mainly combine known ideas (trajectory recombination, learned dynamics, and model-based scoring). The paper could clarify theoretical contributions beyond engineering integration — e.g., formal guarantees, diversity metrics, or physical constraint satisfaction proofs.

Both Task 1 (adapting to diverse human behaviors) and Task 2 (generalizing to unseen physical attributes) require human demonstrations or real-time human interaction. Although the authors introduce AI-AI self-play as a proxy evaluation, the true assessment of human-AI coordination still depends on human testing, which is labor-intensive and inconsistent.

Each human-in-the-loop evaluation requires participants to manually control agents, making experiments costly and variable. Behavioral differences among participants can significantly affect results, reducing reproducibility and preventing large-scale or cross-lab benchmarking.

**Questions:**

See Weakness.

---

> ### Author Response · Authors · 2025-11-24
>
> We thank Reviewer EuNa for the feedback on our work. We are glad to see
> that the reviewer acknowledges the significance of Moving Out as a
> bridge between the symbolic/task-based planning and physically grounded
> controls, the strengths of the BASS framework, and the clarity of our
> empirical analyses. We address the remaining questions in detail below.
>
> ## W1 Contributions in the method and theoretical aspects.
>
> We would like to clarify that our method is not a simple combination of
> known techniques that are applied to single agents. Performing
> trajectory augmentation, simulation, and action selection in a
> multi-agent collaborative environment requires solving challenges that
> do not appear in single-agent settings. Here, we highlight the
> differences between a single-agent extension of existing methods and our
> multi-agent design.
>
> **Behavior Augmentation**: In collaborative manipulation
> (co-manipulation), both agents simultaneously influence the shared
> objects, and their behaviors are tightly coupled. This indicates that
> the start and the end of sub-trajectories must be well-aligned, i.e.,
> satisfying the boundary conditions of co-manipulation, and both agents
> must maintain consistent goals. When augmenting behavior data through
> recombination, a trajectory segment that appears valid for one agent
> cannot be swapped arbitrarily with this agent's trajectories in other
> demonstrations because its partner's actions may no longer form a
> coherent joint behavior after recombination. A segment is
> interchangeable only when the corresponding behaviors in the two
> demonstrations are compatible, such as when an agent begins from nearly
> the same state and pursues the same intermediate objective in both
> segments. Our behavior augmentation follows this condition to preserve
> the intended coordination pattern and yield a new trajectory that
> remains feasible. We have added Appendix O.1 in the revised version to
> show a visualization example.
>
> To better demonstrate this difference, we compare the multi-agent
> recombination in BASS with a single-agent recombination baseline that
> ignores partner alignment.
> The table below shows that single-agent recombination
> increases diversity metrics, but it reduces collaboration performance
> because the generated trajectories no longer represent valid joint
> behaviors. This validates that multi-agent alignment introduced in BASS
> is essential.
> | Method                         | DTW Mean | DTW Var | Entropy (KDE) | Coverage (RBF) | TCR ↑  | NFD ↑  | WT ↓   | AC ↑   |
> |--------------------------------|----------|---------|----------------|----------------|--------|--------|--------|--------|
> | **Multi-agent Recombination (Ours)** | 7.526    | 6.612   | 0.892          | 0.910          | **0.403** | **0.511** | **0.308** | **0.840** |
> | **Single-agent Recombination**       | **7.741** | **6.722** | **0.897**        | **0.919**        | 0.368  | 0.451  | 0.338  | 0.839  |
> | **No Recombination**                 | 7.013    | 6.065   | 0.888          | 0.899          | 0.383  | 0.482  | 0.345  | 0.824  |
>
>
>
>
> **Action Simulation and Selection** Similarly, for the action simulation
> and selection module, if the dynamics model considers only one agent's
> action without its partner, the improvement nearly disappears.
> The table below shows that shows that removing partner information
> yields performance close to not using simulation at all. This again
> confirms that our multi-agent model structure is necessary and that our
> method cannot be reduced to the known single-agent components.
> | Method                             | TCR ↑    | NFD ↑    | WT ↓     | AC ↑     |
> |------------------------------------|----------|----------|----------|----------|
> | **Multi-agent Action Simulation (Ours)** | **0.420** | **0.554** | **0.310** | **0.848** |
> | **Single-agent Action Simulation**       | 0.319    | 0.458    | 0.327    | 0.833    |
> | **No Action Simulation**                 | 0.313    | 0.452    | 0.335    | 0.821    |
>
>
> These results show that our approach can handle multi-agent coordination
> constraints and is not a direct reuse of single-agent ideas. Although a
> full theoretical guarantee is outside the scope of this work, we
> appreciate the suggestion and have included theoretical
> characterizations in the discussion of the future work.

---

> > ### Author Response · Authors · 2025-11-24
> >
> > ## W2, W3 Reliance on human evaluations.
> >
> > We agree that human-in-the-loop experiments require significant human
> > effort. However, this is an inherent and irreducible aspect of the
> > human-AI collaboration research domain where AI must work with real
> > humans. To reduce iterative human involvement during model development
> > and improve reproducibility, prior works have been using AI agents as a
> > proxy before running human evaluations. This is a common practice in
> > human-AI collaboration research. Works such as \[1\], \[2\], and \[3\]
> > follow the same evaluation paradigm. Besides adopting the existing
> > evaluation protocol, our proxy evaluation also includes new ideas to
> > ensure that the evaluation in generalization to unseen behavior and
> > physical attributes is reproducible.
> >
> > In Task 1, previous work usually trains one policy on all human
> > demonstrations as a human proxy. Instead, we split the human data into
> > multiple subsets and run repeated cross-partner testing. This gives a
> > more systematic and comprehensive evaluation of performance with unseen
> > human partners. We designed this protocol so that researchers can easily
> > reproduce the results and run large-scale tests. We also improve
> > reproducibility in Task 2. During data collection, the physical
> > parameters are randomized. During evaluation the maps and settings are
> > fixed. This allows the task to be repeated reliably across runs.
> >
> > \[1\] Dizdarević, Tin, et al. \"Ad-Hoc Human-AI Coordination
> > Challenge.\" ICML 2025
> >
> > \[2\] Liang, Yancheng, et al. "Learning to cooperate with humans using
> > generative agents." NeurIPS 2024
> >
> > \[3\] Yu, Chao, et al. "Learning zero-shot cooperation with humans,
> > assuming humans are biased." ICLR 2023
> >
> > In addition to the new evaluation protocol with the proxy AI agents, we
> > also provide a protocol to reproduce the user study. In our experiments,
> > we conduct user study after we finish the proxy evaluation. We recruited
> > 32 participants and let each person play multiple rounds. This reduces
> > the effect of individual differences and makes the results more stable.
> > We also released the details of evaluation procedures of the user study
> > to support reproducibility. We believe this provides a solid and
> > reliable evaluation of human-AI collaboration.
> >
> > We hope that our responses address the reviewers' concerns, and we would
> > be very happy to provide any further clarification or discussion if
> > needed.

---

### Meta-Review · Area_Chair_suko · 2025-12-22

**Summary:**

The submission introduces a human-AI collaboration benchmark that resembles a range of collaboration modes affected by physical attributes and constraints.  Reviewers liked the idea but found the design of the two tasks too limited and the evaluation lacking control and reproducibility.  There are other concerns about the presentation and its relationship to other existing benchmarks.

**Reviewer Concerns:**

The rebuttal addressed the control of the evaluation; however, concerns about the limited number of tasks and the physical grounding in the real world seem difficult to address.

**Reviewer Scores:**

Reviewers will most likely keep their scores of 6, 6, 4.

---

### Decision · Program_Chairs · 2026-01-26

Reject